



# Inferring potential landslide damming using slope stability, geomorphic constraints and run-out analysis; case study from the NW Himalaya

## Vipin Kumar[1*], Imlirenla Jamir[2], Vikram Gupta[3], Rajinder K. Bhasin[4]

[1]Georisks and Environment, Department of Geology, University of Liege, Liege, Belgium
[2]Public Works Department (PWD), Nagaland, India
[3]Wadia Institute of Himalayan Geology, Dehradun, India
[4]Norwegian Geotechnical Institute, Oslo, Norway

*Correspondence: v.chauhan777@gmail.com; B-18, B-4000, Sart-Tilman, Liege, Belgium

**ABSTRACT**
Prediction of potential landslide damming has been a difficult process owing to uncertainties
related to landslide volume, resultant dam volume, entrainment, valley configuration, river
discharge, material composition, friction and turbulence associated with material. In this
study instability pattern of landslides, parametric uncertainty, geomorphic indices, post
failure run-out predictions and spatio-temporal pattern of rainfall and earthquake is explored
using Satluj valley, North-West (NW) Himalaya as a case study area to predict the potential
landslide damming sites.  The study area witnessed landslide damming in the past and
incurred $ ~30M loss and 350 lives in the last four decades due to such processes. Forty four
active landslides in the study area that cover total ~$4.81 \pm 0.05 \times 10^6 \, m^2$ area and ~$34.1 \pm 9.2$
x $10^6 \, m^3$ volume are evaluated in the study to identify those that may result in potential
landslide damming. Out of forty four, five landslides covering the volume of ~$26.3 \pm 6.7$ x
$10^6 \, m^3$ are observed to form potential landslide dams. Spatio-temporal varying pattern of
rainfall in recent years enhances the possibility of landslide triggering and hence potential
damming. These landslides also resulted in $24.8 \pm 2.7$m to $39.8 \pm 4.0$m high material flow in
run-out predictions.
**Key words:** Landslide damming, Slope stability; Run-out; Himalaya



## 1.0 INTRODUCTION

Landslide damming is a normal geomorphic process in narrow valleys and has been one of most disastrous natural processes (Dai et al. 2005; Gupta and Sah 2008; Delaney and Evans 2015). There have been many studies that explored the damming characteristics (Li et al. 1986; Costa and Schuster 1988; Takahashi and Nakawaga 1993; Ermini and Casagli 2003; Fujisawa et al. 2009; Stefanelli et al. 2016; Kumar et al. 2019a). However, studies concerning the prediction of potential landslide dams and their stability at regional scale have been relatively rare, particularly in Himalaya despite a history of landslide damming and flash floods (Gupta and Sah 2008; Ruiz-Villanueva et al. 2016; Kumar et al. 2019a). In order to identify the landslides that have potential to form dams, following factors have been main requisites; (i) pre- and post-failure behavior of landslide slopes (ii) landslide volume, stream power and morphological setting of the valley.

To understand the pre-failure pattern, slope stability evaluation through numerical modeling has been a common practice. The Finite Element Method (FEM) has been a most widely used numerical model for the complex slope geometry (Griffiths and Lane 1999; Jing 2003; Kanungo et al. 2013; Jamir et al. 2017; Kumar et al. 2018). However, selection of input parameters in the FEM analysis and set of assumptions (material model, failure criteria, convergence etc.) may also result into uncertainty in final output (Wong 1984; Cho 2007; Li et al. 2016; Siddique and Pradhan 2018). Input parameters based uncertainty can be resolved by performing the sensitivity/parametric analysis and utilization of most appropriate criteria can minimize the uncertainty caused by assumptions. Post-failure behavior of landslides can be understood through run-out analysis (Hungr et al. 1984; Hutter et al. 1994; Rickenmann and Scheidl 2013). These methods could be classified into empirical/statistical and dynamical categories (Rickenmann 2005). Owing to flexibility in rheology, solution approach, reference frame, and entrainment, dynamic models have been relatively more realistic for site-specific problems (Corominas and Mavrouli 2011). Though different numerical models have different advantages and limitations, Voellmy rheology (friction and turbulence) (Voellmy 1955; Salm 1993) based Rapid Mass Movement Software (RAMMS) (Christen et al. 2010) model have been used widely owing to inclusion of rheological and entrainment rate flexibility.

Apart from pre-and post-failure pattern, landslide volume, stream power and morphological setting of the valley are crucial to infer potential landslide damming. Morphological Obstruction Index (MOI) and Hydro-morphological Dam Stability Index (HDSI) have been





most widely used geomorphic indices involving landslide volume, stream power and the
morphological setting of valley to infer the potential of landslide dam formation and their
temporal stability (Costa and Schuster 1988; Ermini and Casagli 2003; Stefanelli et al. 2016).
The NW Himalaya, India has been a most affected terrain by the landslides owing to active
tectonics and seasonal precipitation sources i.e., Indian Summer Monsoon (ISM) and Western
Disturbance (WD). The WD has been southward extension of sub-tropical westerly jet (Dimri
et al. 2015). The NW Himalaya has accommodated ~51 % of all the landslides in
India during yrs. 1800-2011 (Parkash 2011). The Satluj River valley, NW Himalaya is one
such region that has claimed ~350 lives and loss of minimum 30 million USD due to the
landslides and associated floods in the last four decades and holds a high potential for
landslide damming and resultant floods (Ruiz-Villanueva et al. 2016; Kumar et al.
2019a). Therefore, Satluj River valley is taken as a case study area, of which 44 landslides
(20 debris slides, 13 rockfalls, and 11 rock avalanches) belonging to different litho-tectonic
regimes are modeled using the FEM technique. Multiple slope sections and a range of values
of different input parameters are used to perform parametric study.  In order to determine the
human population that may get affected by these landslides, census statistics are also used.
Morphological obstruction index and Hydro-morphological dam stability index are used to
determine the potential of landslide dam formation, if failure occurs, and their stability in
case of formation. In view of the role of rainfall and earthquake as main landslide triggering
factors, spatio-temporal regime of these two factors in the study area is also discussed. Run-
out prediction of certain landslides using the RAMMS model is also performed to understand
their contribution in potential landslide damming. This study provides detailed insight into
regional instability pattern, associated uncertainty, and potential landslide damming sites and
hence can be replicated in other hilly terrain witnessing frequent landslides and damming.
**2.0 STUDY AREA**
The study area is located between the Moorang (31°36′1″ N, 78°26′ 47″ E) and Rampur town
(31°27′10″ N, 77°38′ 20″ E) of the Satluj River valley, NW Himalaya (Fig. 1). The Satluj
River flows across Tethyan Sequence (TS), Higher Himalaya Crystalline (HHC), Lesser
Himalaya Crystalline (LHC), and Lesser Himalaya Sequence (LHS). The TS in the study area
comprises slate/phyllite and schist and has been intruded by the biotite-rich granite i.e.,
Kinnaur-Kailash Granite (KKG) near the Sangla Detachment (SD) fault (Sharma 1977;
Vannay et al. 2004). The SD fault separates the TS from the underlying crystalline rockmass



of the HHC. Migmatitic gneiss marks the upper part of the HHC whereas, base is marked by
kyanite-sillimanite gneiss rockmass (Sharma 1977; Vannay et al. 2004; Kumar et al. 2019b).
The Main Central Thrust (MCT) fault separates the HHC from the underlying schist/gneissic
rockmass of the LHC. The LHC comprises mica schist, carbonaceous schist, quartzite, and
amphibolite. A thick zone of gneiss i.e., Wangtu Gneissic Complex (WGC) is also exposed in
the LHC, which comprises augen gneiss and porphyritic granitoids.  The LHC is delimited at
the base by the Munsiari Thrust (MT) fault that is thrusted over the Lesser Himalaya
Sequence (LHS) rockmass. The MT contains breccia, cataclastic, and fault gouge (Sharma
1977; Vannay et al. 2004; Kumar et al. 2019b). The LHS in the study area consists of quartz-
arenite (Rampur Quartzite) with bands of phyllite, meta-volcanics, and paragneiss (Sharma

92   1977).

The present study covers forty-four (44) active landslides (20 debris slides, 13 rock falls, and
11 rock avalanches) along the study area that have been mapped recently by Kumar et al.
(2019b). The location and dimensional details of the landslides have been summarized in
Table 1. Field photographs of few of these landslides are presented in Fig. 2. The TS  and
LHS in the study area has been subjected to tectonic tranquility with exhumation rates as low
as 0.5 - 1.0 mm/yr whereas, the HHC and LHC region comprise 1.0 - 4.5 mm/yr rate of
exhumation (Thiede et al. 2009; Kumar et al. 2019b). The MCT fault region and the WGC
are noted to have maximum exhumation rate (i.e., ~4.5 mm/yr) that is evident from the deep
gorges in these regions (Fig. 2). Further, a majority of the earthquake events in the study area
in the last 7 decades have been related to the N-S oriented Kaurik - Chango Fault (KCF) that
is subjected to the Karakoram Fault (KF) (Kundu et al. 2014; Hazarika et al. 2017;
International Seismological Centre 2019). The climate zones in the study area also show
spatial variation from humid (~800 mm/yr) in the LHS to semi-arid (~200 mm/yr) in the TS
(Kumar et al. 2019b). The HHC acts as a transition zone where climate varies from semi-
humid to semi-arid in SW-NE direction. This transition has been attributed to the 'orographic
barrier' nature of the HHC that marks the region in its north as orographic interior and the
region to its south as the orographic front (Wulf et al. 2012; Kumar et al. 2019b).
The landslides in the study area have been a consistent threat to the socio-economic condition
of the nearby human population (Gupta and Sah 2008; Ruiz-Villanueva et al. 2016; Kumar et
al. 2019a). Therefore, the human population in the vicinity of each landslide was also
determined by considering villages/town in that region. It is to note that total 25,822 people
reside in the 500 m radius of 44 landslide slopes and about 70 % of the this population is



residing in the reach of debris slide type landslides. Since the Govt. of India follows a 10 year
gap in census statistics, the human population data was based on last official i.e., Census-
2011. The next official census is due in 2021. It is to note that the exact population in year
2020-2021 might be higher than that of census 2011 that would be reflected in census 2021.
**3.0 METHODOLOGY**
In order to determine the potential landslide sites along the Satluj River valley, NW
Himalaya, methodology involved field data collection, satellite imagery analysis, laboratory
analyses, Finite Element Method (FEM) bases slope stability evaluation, parametric analysis,
application of Morphological Obstruction Index (MOI) & Hydro-morphological Dam
Stability Index (HDSI) and debris run-out analysis. Details are as follows;
*3.1 Field data, satellite imagery processing and laboratory analyses*
The field work involved rock/soil sample collection from each landslide location, rockmass
joint mapping, and N-type Schmidt Hammer Rebound (SHR) measurement. The joints were
included in the slope model for the FEM analysis. Dataset involving the joint details is
uploaded to the open accessed *Mendeley Data* repository (Kumar et al. 2020). The SHR
values were obtained as per International Society of Rock Mechanics (ISRM) standard
(Aydin 2008).
The Cartosat-1satellite imagery and field assessment were used to finalize the location of
slope sections (2D) of the landslides. The Cartosat-1 imagery has been used widely for
landslide related studies (Martha et al. 2010).  The Cartosat-1Digital Elevation Model
(DEM), prepared using the Cartosat-1 stereo imagery, was used to extract the 2D slope
sections of the landslides using Arc GIS-10.2 software. Details of the satellite imagery are
mentioned in Table 2.
Rock/soil samples were analyzed in the National Geotechnical Facility (NGF) and Wadia
Institute of Himalayan Geology (WIHG) laboratory, India. The rock samples were drilled and
smoothened for Unconfined Compressive Strength (UCS) (IS: 9143-1979) and ultrasonic test
(CATS Ultrasonic (1.95) of Geotechnical Consulting & Testing Systems. The Ultrasonic test
was conducted to determine the density, elastic modulus, and Poisson's ratio of rock samples.
The soil samples were tested for grain size analysis (IS: 2720-Part 4-1985), UCS test (IS:
2720-Part 10-1991), and direct shear test (IS: 2720-Part 13- 1986). If the soil samples
contained < 5% fines (< 75 mm), hydrometer test was not performed for the remaining fine



material. In the direct shear test, soil samples were sheared under constant normal stress of
50, 100 and 150 kN/m$^2$. The UCS test of soil was performed under three different rates of
movements i.e., 1.25 mm/min, 1.50 mm/min and 2.5 mm/min.
*3. 2 Slope stability and parametric analyses*
The Finite Element Method (FEM) was performed along with the Shear Strength Reduction
(SSR) technique to infer the critical Strength Reduction Factor (SRF), Shear Strain (SS), and
Total Displacement (TD) in the 44 landslide slopes (20 debris slides, 13 rock falls, and 11
rock avalanche) using the RS2 software. The SRF has been observed to be similar in nature
as the Factor of Safety (FS) of the slope (Zienkiewicz et al. 1975; Griffiths and Lane 1999).
To define the failure in the SSR approach, non-convergence criteria was used (Nian et al.
2011). The boundary condition with the restraining movement was applied to the base and
back, whereas the front face was kept free for the movement (Fig.3). In-situ field stress was
adjusted in view of dominant forces i.e., extension or compression by changing the value of
the coefficient of earth pressure (k). The $k = \sigma_h/\sigma_v = 0.5$ was used in extensional regime,
whereas $k = \sigma_h/\sigma_v = 1.5$ was used in compressional regime. The spatial variability of
compressional and extensional regime in this collisional orogeny region has been discussed in
detail by Vannay et al. (2004).
The soil and rock mass were used in the FEM analysis through Mohr-Coulomb (M-C) failure
criterion (Coulomb 1776; Mohr 1914) and Generalized Hoek-Brown (GHB) criterion (Hoek
et al. 1995), respectively. The parallel- statistical distribution of the joints with normal-
distribution joint spacing in the rock mass was applied through Barton-Bandis (B-B) slip
criterion (Barton and Choubey 1977; Barton and Bandis 1990). Plane strain triangular
elements having 6 nodes were used through the graded mesh in the models. Details of the
criteria used in the FEM analysis are mentioned in Table 3. Dataset involving the value of
input parameters used in the FEM analysis is uploaded to the open accessed *Mendeley Data*
repository (Kumar et al. 2020). It is to note that the FEM analysis is performed under static
load i.e., field stress and body force. The dynamic analysis is not performed, at present, in
absence of any major seismic events in the region in last 4 decades (sec. 4.4) and lack of
reliable dynamic load data of nearby major seismic events.
To understand the uncertainty caused by the selection of 2D slope section, multiple slope
sections were taken, wherever possible. More than one slope sections were modeled for each
debris slide, whereas for rock falls/ rock avalanche only one slope section could be chosen





due to the limited width of the rock falls/rock avalanche in the study area. To find out the
relative influence of different input parameters on the final output, a parametric study was
also performed. In the parametric study for debris slides, Akpa landslide (S.N.5 in Fig. 1),
Pangi landslide (S.N.13 in Fig. 1), and Barauni Gad landslide (S.N.38 in Fig. 1) were chosen,
whereas Tirung khad (S.N.2 in Fig.1) and Chagaon landslide (S.N.21 in Fig. 1) were
considered to represent rock fall. Baren Dogri (S.N.7 in Fig. 1) landslide was used to
represent rock avalanches. The configuration of these landslide models is presented in Fig. 3.
The selection of these landslides for parametric study was based on following two factors; (1)
to choose landslides from different litho-tectonic regime, (2) representation of varying stress
regime i.e., extensional, compressional, and relatively stagnant. The Parametric study of the
debris slide models involved following 9 parameters; field stress coefficient, stiffness ratio,
cohesion and angle of friction of soil, elastic modulus and Poisson's ratio of soil, rockmass
modulus, Poisson's ratio and uniaxial compressive strength of rock. For the rockfalls/rock
avalanche, following 6 parameters; uniaxial compressive strength of rock, rockmass modulus
of rock, Poisson's ratio of rock, 'mi' parameter, stiffness ratio, and field stress coefficient
were used.  The 'mi' is a Generalized Hoek-Brown (GHB) parameter that is equivalent to the
angle of friction of Mohr-coulomb (M-C) criteria.
*3. 3 Potential landslide dam formation & stability evaluation*
Considering the possibility of landslide dam formation in case of slope failure, following
geomorphic indices are also used;

(i)      Morphological Obstruction Index (MOI)

$MOI= \log (V_l/W_v)$                    **Eq. 1**

(ii)     Hydro-morphological Dam Stability Index (HDSI)

$HDSI= \log (V_d/A_b.S)$                 **Eq. 2**

Where, $V_d$ (dam volume)= $V_l$ (landslide volume), $m^3$; $A_b$ is upstream catchment area ($km^2$);
$W_v$ is width of dammed valley (m) and S is local slope gradient of river channel (m/m).
Though the resultant dam volume could be higher or lower than the landslide volume owing
to slope entrainment, rockmass fragmentation, retaining of material at the slope, and washout
by the river (Hungr and Evans 2004; Dong et al. 2011), dam volume is assumed to be equal



to landslide volume for worst case. By utilizing the comprehensive dataset of ~300 landslide
dams of Italy, Stefanelli et al. (2016) have classified the MOI into (i) non-formation domain:
MOI <3.00 (ii) uncertain evolution domain: 3.00 <MOI >4.60 and (iii) formation domain:
MOI >4.60. Similarly, utilizing the same dataset, Stefanelli et al. (2016) defined the HDSI
into following categories (i) instability domain: HDSI <5.74 (ii) uncertain determination
domain: 5.74<HDSI >7.44 and (iii) Stability domain: HDSI>7.44.
*3. 4 Rainfall and Earthquake regime*
Precipitation in the study area owes its existence to Indian Summer Monsoon (ISM) and
Western Disturbance (WD) and varies spatially-temporally due to various localized and
external factors (Gadgil et al. 2007; Hunt et al. 2018). Therefore, we have taken
TRMM_3B42 daily rainfall data of year 2000-2019 at four different locations; Moorang (in
Tethyan Sequence), Kalpa (in Higher Himalaya Crystalline), Nachar (in Lesser Himalaya
Crystalline) and Rampur (in Lesser Himalaya Sequence). The dataset of earthquake events
(2<M<8) in and around study area during year 1940-2019 was retrieved from International
Seismological Centre (ISC) catalogue (http://www.isc.ac.uk/iscbulletin/search/catalogue/) to
determine the spatio-temporal pattern.
*3. 5 Run-out analysis*
Since the study area has witnessed many disastrous landslides, mostly rainfall triggered, and
flash floods in past (Gupta and Sah 2008; Ruiz-Villanueva et al. 2016), run-out analysis was
carried out to understand the post-failure scenario. Such run-out predictions will also be
helpful to ascertain the possibility of damming because various studies have observed the
river damming by debris flows (Li et al. 2011; Braun et al. 2018). Therefore, the landslides
those have potential to form the landslide dams based on indices analysis (sec. 3.3) are
evaluated for such run-out analysis.
In this study, Voellmy rheology (Voellmy 1955; Salm 1993) based Rapid Mass Movement
Software (RAMMS) (Christen et al. 2010) model is used to understand the run-out pattern.
The RAMMS for debris flow uses the Voellmy friction law and divides the frictional
resistance into a dry-Coulomb type friction ($\mu$) and viscous-turbulent friction ($\xi$). The
frictional resistance S (Pa) is thus;
$$S = \mu N + (\rho g u^2)/\xi \qquad\qquad \text{Eq. 3}$$



where $N$; $\rho$hgcos($\phi$) is the normal stress on the running surface, $\rho$; density, g; gravitational
acceleration, $\phi$; slope angle, h; flow height and u= (ux, uy), consisting of the flow velocity in
the x- and y-directions. In this study, a range of friction ($\mu$) and turbulence ($\xi$) values, apart
from other input parameters, are used to eliminate the uncertainty in output (Table 4).
Generally, the values for $\mu$ and $\xi$ parameters are achieved using the reconstruction of real
events through simulation and subsequent comparison between dimensional characteristics of
real and simulated event. However the landslides in the study area merge with the river floor
and/or are in close proximity and there is no failed material left from previous events to
reconstruct. Therefore, $\mu$ and $\xi$ values were taken in a range in view of topography of
landslide slope and run-out path, landslide material, similar landslide events/material and
based on previous studies/models (H¨urlimann et al. 2008; Rickenmann and Scheidl 2013;
RAMMS v.1.7.0). Since these landslides are relatively deep in nature and we are of
understanding that during slope failure, irrespective of type of trigger, entire loose material
might not slide down, the depth of landslide is taken as only ¼ (thickness) in the run-out
calculation.

## 4.0 RESULTS

**4.0 RESULTS**
*4.1 Slope instability regime and parametric output*
Results indicated that out of 44 landslides (20 debris slides, 13 rockfalls, and 11 rock
avalanches), 31 are in meta-stable state (1 $\leq$FS$\leq$ 2) and 13 in unstable state (FS <1) (Fig. 4).
Most of the unstable landslides are debris slides, whereas the majority of the meta-stable
landslides (1 $\leq$FS$\leq$ 2) are rock fall/rock avalanche. Debris slides constitute ~ 90 % and ~99 %
of the total area and volume, respectively of the unstable landslides. It is to note that about
~70 % of the total human population along the study area resides in the vicinity (~500 m) of
these unstable debris slides (Fig. 4). Rock falls/Rock avalanches constitute ~84 % and ~78 %
of the area and volume, respectively of the meta-stable landslides. Out of total twenty debris
slides, twelve debris slides are found to be in unstable (FS <1) stage whereas, eight in meta-
stable condition (1 <FS< 2) (Fig. 4). These twenty debris slides occupy ~1.9 ±0.02 x $10^6$ $m^2$
area and ~ 26 ±6 x $10^6$ $m^3$ volume. While comparing the factor of safety with the Total
Displacement (TD) and Shear Strain (SS), nonlinear poor correlation is achieved (Fig. 5).
Since, the TD and SS present a relatively good correlation (Fig. 5), only the TD is used
further alongwith the FS. The TD ranges from 7.4± 8.9 cm to 95.5± 10 cm for unstable debris
slides and ~18.8 cm for meta-stable landslides (Fig. 4).



Out of thirteen (13) rockfalls, one (1) belongs to the unstable state (FS <1) and twelve (12) to
the meta-stable state (1 <FS< 2) (Fig. 4). The TD varies from 0.4 to 8.0 cm with the
maximum for Bara Kamba rockfall (S.N. 31 in Fig. 1) in the Lesser Himalaya Crystalline.
Out of eleven (11) rock avalanches, one (1) belongs to the unstable state (FS <1) and ten (10)
to the meta-stable state (1<FS<2) (Fig. 4). The TD varies from 6.0 to 132.0 cm with the
maximum for the Kandar rock avalanche (S.N. 25 in Fig. 1) of the Lesser Himalaya
Crystalline. It is noteworthy that relatively higher TD is obtained by the rock fall and rock
avalanche of the Lesser Himalaya Crystalline region (Fig. 4). The landslides of the Higher
Himalaya Crystalline (HHC), Kinnaur Kailash Granite (KKG) and Tethyan Sequence (TS),
despite being only 17 out of the total 44 landslides, constituted ~ 67 % and ~ 82 % of the total
area and total volume of the landslides.
The Factor of Safety (FS) of debris slides is found to be relatively less sensitive to the change
in the value of input parameters than the Total Displacement (TD) (Fig. 6). In case of Akpa
('a' in Fig. 6) and Pangi landslide ('b' in Fig. 6), soil friction and field stress have more
influence on the FS. However, for the TD, field stress, elastic modulus and Poisson's ratio of
soil are relatively more controlling parameters. The FS and TD of the Barauni Gad landslide
('c' in Fig. 6) are relatively more sensitive to soil cohesion and 'mi' parameter. Therefore, it
can be inferred that the FS of debris slides is more sensitive to soil friction and field stress,
whereas TD is mostly controlled by field stress and deformation parameters i.e, elastic
modulus and Poisson's ratio. Similar to the debris slides, the FS of rock falls and rock
avalanche are found to be relatively less sensitive than TD to the change in the value of input
parameters (Fig. 7). In case of Chagaon rock fall ('c' in Fig. 7), poission ratio and UCS have
relatively more influence on FS and TD. Tirung Khad rock fall ('a' in Fig. 7) and Baren
Dogri rock avalanche ('b' in Fig. 7) show dominance of 'mi' parameter and field stress in the
FS as well as in TD. Thus, it can be inferred that the rock fall/rock avalanche are more
sensitive to 'mi' parameter and field stress.
*4.2 Potential landslide damming*
Based on the MOI, out of total 44 landslides, 5 (S.N. 5, 7, 14, 15, 19) are observed to be in
formation domain, 15 in uncertain domain and 24 in non-formation domain, at present (Fig.
8a). These five landslides that have potential to dam the river in case of slope failure
accommodate ~26.3 ± 6.7 x $10^6$ m$^3$ volume (Fig. 8). The five landslides are also presented
separately in Fig. 9 (a-e).



In terms of temporal stability (or durability), out of five landslides that have potential to block
the river, only one landslide (S.N. 5) is noted to attain the uncertain domain, whereas
remaining four show instability (Fig. 8b,d). The lacustrine deposit in the upstream of Akpa
landslide (S.N. 5) in Fig. 9a implies the signs of landslide damming in the past too (Fig. 10).
The uncertain temporal stability indicates that the landslide dam may be stable or unstable
depending upon the stream power and landslide volume, which in turn are dynamic factors
and may change owing to changing climate and/or tectonic event. The landslides that have
been observed to form the landslide dam but are noted to be in temporally unstable category
(S.N. 7, 14, 15, 19) are still considerable owing to associated risks of lake-impoundment and
generation of secondary landslides. Urni landslide (S.N. 19) (Fig. 9e) that damaged the part
of National Highway road (NH)-05 has already partially dammed the river since year 2016
and holds potential for further damming (Kumar et al. 2019a). Apart from S.N. 5 and S.N. 19
landslides, remaining landslides (S.N. 7, 14, 15 in Fig. 1) belong to Higher Himalaya
Crystalline (HHC) region that has been observed to accommodate many landslide damming
and subsequent flash floods events in the past (Sharma et al. 2017).
*4.3 Rainfall and Earthquake regime*
In order to explain the spatio-temporal variation in rainfall, topographic profile of the study
area is plotted along with the rainfall variation (Fig. 11a). The temporal distribution of
rainfall is presented at annual, monsoonal (SW Indian Monsoon: June-September) and non-
monsoonal (Western Disturbance: Oct-May) level (Fig. 11b-d). Rainfall data of year 2000-
2019 revealed a relative increase in annual rainfall since year 2010 (Fig. 11b). The Kalpa
region (situated in orographic barrier setting) received relatively more annual rainfall than the
Rampur, Nachar and Moorang region throughout the time period, except year 2017. The
rainfall dominance at Kalpa is more visible in non-monsoonal season (Fig. 11d). It may be
due to its orographic influence on the saturated winds of western disturbance. Further, the
rainfall during the monsoon  season that was dominant at the Rampur region till year 2012
gained dominance at Kalpa region since year 2013 (Fig. 11c).
Extreme rainfall events of June 2013 that resulted in widespread slope failure in the NW
Himalaya also caused landslide damming at places (NDMA 2013; Kumar et al. 2019a).
Similar to the year 2013 rainfall event, the year 2007, 2010 and 2019 also witnessed
enhanced annual rainfall and associated flash floods and/or landslides in the region
(hpenvis.nic.in, retrieved on March 1, 2020; sandrp.in, retrieved on March 1, 2020).





However, the contribution of ISM season rainfall and WD associated rainfall has been
variable in these years (Fig. 11). Such frequent but inconsistent rainfall events that possess
varied (temporally) dominance of ISM and WD are observed to owe their occurrence to
following local and regional factors; El-Nino Southern Oscillation (ENSO), Equatorial Indian
Ocean Circulation (EIOC) and planetary warming (Gadgil et al. 2007; Hunt et al. 2018).
Orographic setting is noted to act as principle local factor as evident from relatively more
rainfall (total precipitation=1748±594 mm/yr.) at Kalpa region (orographic barrier) in the
non-monsoon and monsoon season from the year 2010 onwards (Fig. 11). Prediction of
potential landslide damming sites in the region revealed that four (S.N. 7, 14, 15, 19) out of
five landslides that can form the dam belong to this orographic barrier. Therefore, in view of
the prevailing rainfall trend since the year 2010, regional factors, discussed above, and
orographic setting, precipitation triggered slope failure events cannot be denied in the future.
Such slope failure events, if occurred, at the predicted landslide damming sites may certainly
dam the river.
The seismic pattern revealed that the region has been hit by 1662 events with epicenters
located in and around the study area (Fig. 12a). However, ~99.5 % of these earthquake events
had a magnitude of less than 6.0 and only 8 events are recorded in the range of 6.0 to 6.8 $M_s$
(International Seismological Centre 2019). Out of these 8 events, only one event i.e., 6.8 $M_s$
(19[th] Jan. 1975) has been noted to induce widespread slope failures in the study area (Khattri
et al. 1978). The majority of the earthquake events in the study area has occurred in the
vicinity of the N-S oriented trans-tensional Kaurik - Chango Fault (KCF) that accommodated
the epicenter of 19[th] Jan. 1975 earthquake (Hazarika et al. 2017; International Seismological
Centre 2019). It is to note that about 95% of total 1662 events had their focal depth within 40
km (Fig. 12b). Such a relatively low magnitude - shallow seismicity in the region has been
related to the Main Himalayan Thrust (MHT) decollement as a response to relatively low
convergence (~14±2 mm/yr) of India and Eurasia plates in this region (Bilham 2019) (Fig.
12c). Further, the arc (Himalaya)-perpendicular Delhi-Haridwar ridge that is underthrusting
Eurasian plate in this region has been observed to be responsible for the spatially varied *low*
seismicity in the region (Hazarika et al. 2017). Thus, though the study area has been
subjected to low magnitude-shallow seismicity, chances of earthquake-triggered landslides
have been relatively low in comparison to rainfall-triggered landslides and associated
landslide damming. For this reason and lack of reliable dynamic load of major earthquake



event, we have performed the *static* modeling in the present study. However, we intend to
perform the *dynamic* modeling in near future if reliable dynamic load data will be available.
*4.4 Run-out analysis*
All five landslides (S.N. 5, 7, 14, 15, 19 in Fig. 8, 9) that are observed to form potential
landslide dam in the event of slope failure were also considered for the run-out analysis.
Results are as follows;
*4. 4.1 Akpa landslide (S.N. 5)*
Though it is difficult to ascertain that how much part of the debris flow might contribute in
the river blockage, it will certainly block the river in view of ~38 m high debris material with
~50 m wide run-out across the channel in this narrow part of river valley (Fig. 9a) even at
maximum value of coefficient of friction (i.e., μ =0.3) (Fig. 13a). It is to note that not only
the run-out extent but flow height also decreases on increasing friction value (Fig. 13a.1-
13.a.3). The maximum friction can take into account the possible resistance by vegetation on
slope and bed-load on river channel. However, apart from the frictional characteristics of run-
out path, saturation of debris flow also controls its dimension and hence consequences like
potential damming. To account the saturation of debris flow, different values of turbulence
coefficient (ξ) were used (Table 4). The resultant flow height (representing 9 sets of debris
flow obtained using μ=0.05, 0.1 and 0.3 and ξ= 100,200 and 300 m/s$^2$) attains its peak value
i.e., 39.8± 4.0m at the base of central part of landslide (Fig. 14a).
*4.4.2 Baren dogri landslide (S.N. 7)*
At the maximum friction value (μ =0.4), Baren dogri landslide is noted to attain peak value of
flow height i.e., ~30 m at the base of central part of landslide (Fig. 13b). Similar to the valley
configuration around the Akpa landslide (sec 4.4.1), river valley attains narrow/deep gorge
setting here also (Fig. 9b). The maximum value of debris flow height obtained using different
μ and ξ values is 25.6 ± 2.1m (Fig. 14b). Flow material is also noted to attain more run-out in
upstream direction of river (~1100 m) than in the downstream direction (~800 m). This
spatial variability in run-out length might exist due to river channel configuration as river
channel in upstream direction is relatively narrower than the downstream direction.
*4.4.3 Pawari landslide (S.N. 14)*





Pawari landslide attains maximum flow height of ~20 m at the maximum friction of run-out
path (μ=0.4) (Fig. 13c).The resultant debris flow that is achieved using different values of μ
and ξ parameters attains a peak value of 24.8 ± 2.7 m and decreases gradually with a run-out
of ~1500 m in upstream and downstream direction (Fig. 14c). This landslide resulted in the
relatively long run-out of ~1500 in upstream and downstream direction. Apart from the
landslide volume that affects the run-out extent, valley morphology also controls it as evident
from previous landslides. The river channel in upstream and downstream direction from the
landslide location is observed to be narrow (Fig. 9c).
*4.4.4 Telangi landslide (S.N. 15)*
Telangi landslide is noted to result in peak debris flow height of ~24 m at the maximum
friction (μ=0.4) (Fig. 13d). It is to note that on increasing the friction of run-out path, flow
run-out decreased along the river channel but increased across the river channel resulting into
possible damming. The debris flow after taking into account different values of μ and ξ
parameters attains a peak value of 25.0± 4.0 m (Fig. 14d). Similar to Baren dogri landslide
(S.N. 7), material attained more run-out in upstream direction of river (~1800 m) than in
downstream direction (~600 m) that attributes to narrower river channel in upstream than the
downstream direction. The downstream side attains wider river channel due to the traversing
of Main Central Thrust (MCT) fault in the proximity (Fig. 1). Since Pawari and Telangi
landslide (S.N 14 &15) are situated ~500 m from each other, their respective flow run-outs
might mix in the river channel resulting into disastrous cumulative effect.
*4.4.5 Urni landslide (S.N. 19)*
Urni landslide attained a peak value of ~44 m of debris flow height at the maximum friction
value (μ=0.4) (Fig. 13e). After taking into account different values of μ and ξ parameters, the
debris flow attained a height of 26.3± 1.8 m (Fig. 14e). Relatively wider river channel in
downstream direction (Fig. 9e) is considered to results in long run-out in downstream
direction than in the upstream.
**5.0 DISCUSSION**
Present study aimed to determine the potential landslide damming sites in the Satluj River
valley, NW Himalaya. In order to achieve this objective, 44 landslides along the Satluj River
valley were considered. At first, slope stability evaluation of all slopes was performed
alongwith parametric evaluation. Then geomorphic indices i.e., Morphological Obstruction



Index (MOI) and Hydro-morphological Dam Stability Index (HDSI) were used to predict the formation of potential landslide dam and their subsequent stability. Rainfall and earthquake regime were also explored in the study area. Finally, run-out analysis was performed of those landslides that have been observed to form potential landslide dam.

The MOI revealed that out of forty-four landslides, five (S.N. 5, 7, 14, 15, 19) have potential to form the landslide dam (Fig. 8, 9). On evaluating the stability of such potential dam sites using the HDSI, the landslide (S.N. 5) is noted to attain uncertain domain (5.74<HDSI<7.44) in terms of dam stability. The uncertain term implies that the resultant dam may be stable or unstable depending upon the landslide/dam volume, upstream catchment area (or water discharge) and slope gradient (sec 3.3). Since this landslide presents clear signs of having already formed a dam in the past, as indicated by the alternating fine-coarse layered sediment deposit (or lake deposit) in the upstream region (Fig. 9a, 10), recurrence can't be denied. Further, run-out analysis of landslide has predicted 39.8± 4.0m high debris flow in the event of failure that will block the river completely (Fig. 13a, 14a). However, the durability of the blocking can't be ascertained as it subjected to the volume of landslide that will be retained at the channel and river discharge.

Remaining four landslides (S.N. 7, 14, 15, 19), though showed instability i.e., HDSI <5.74 at present, may form the dam in near future as the region accommodating these landslides has been affected by such damming and subsequent flash floods in the past (Sharma et al. 2017). The last one of these i.e., S.N. 19 (Urni landslide) has already dammed the river partially and holds potential to completely block the river in near future (Kumar et al. 2019a). Run-out analysis of these landslides (S.N. 7, 14, 15, 19) has predicted 25.6 ± 2.1m, 24.8 ± 2.7m, 25.0± 4.0m and 26.3± 1.8m flow height, respectively that will result in temporary blocking of the river (Fig. 13,14). These findings of run-out indicate towards the blocking of river in the event of slope failure, irrespective of durability, despite the conservative depth as input because only ¼ of landslide thickness is used in the run-out analysis (sec. 3.5).

Stability evaluation of these five landslide slopes (S.N. 5, 7, 14, 15, 19) that have potential to form landslide dam revealed that except one landslide (S.N.7) that is meta-stable (1≤FS≤2), at present, remaining four belong to unstable category (FS<1) (Fig. 4). Further, except this landslide that is meta-stable (S.N. 7), remaining four unstable landslide slopes is debris slide in nature. It is noteworthy to discuss the implications of FS<1. The Factor of safety (FS) in the Shear Strength Reduction (SSR) approach is a factor by which the existing shear strength



of material is divided to determine the critical shear strength at which failure occurs
(Zienkiewicz et al. 1975; Duncan 1996). Since the landslide represents a failed slope i.e.,
critical shear strength > existing shear strength, FS<1 is justifiable. Further, the failure state
of a slope in the FEM can be defined by different criteria; the FS of same slope may vary a
little depending upon the usage of failure criteria and convergence threshold (Abramson et al.
1996; Griffiths and Lane 1999).
In general, the possible causes of instability (FS<1) may be steep slope gradient, weak
lithology, and jointed rock mass. Three (S.N. 7, 14, 15) out of these five landslides that have
potential to form the dam belong to the tectonically active Higher Himalaya Crystalline
(HHC) and the notion of steep slope gradient cannot be generalized because the HHC
accommodates most voluminous ($\sim 10^5$-$10^7$ m$^3$) landslides (Fig. 4). These deep seated
landslides must require smooth slope gradient to accommodate the voluminous overburden.
The HHC comprises strong lithology i.e., gneiss therefore, therefore the notion of weak
lithology also may not be appropriate. However, the jointed rock mass that owes its origin to
numerous small scale folds, shearing and faults associated with the active orogeny process
(sec.2.0) can be considered as the main factor for relatively more instability of debris slide
type landslides. Since, the study area is subjected to the varied stress regime caused by the
tectonic structures (Vannay et al. 2004) thermal variations (Singh et al. 2015), and
anthropogenic cause (Lata et al. 2015), joints may continue to develop and destabilize the
slopes. Apart from this inherent factor like joints, external factors like rainfall and
exhumation rate may also contribute to instability of these landslides. This region receives
relatively more annual rainfall owing to orographic barrier setting (Fig. 11) and is subjected
to relatively high exhumation rate of 2.0-4.5 mm/yr (Thiede et al. 2009).
Two landslides (S.N. 5, 19) that are also capable to form potential landslide dam (Fig. 8, 9a;
e) and are also unstable (FS<1) in nature (Fig. 4) do not belong to the HHC. The first
landslide (S.N. 5) exists at the lithological contact of schist and Kinnaur Kailash Granite
(KKG) rockmass and regional normal fault i.e., Sangla Detachment (ST) or South Tibetan
Detachment (STD) passes through this contact. Few studies suggest that the SD normal fault
is an outcome of reactivation of former thrust fault (Vannay et al. 2004) that has resulted in
intense rockmass shearing (Kumar et al. 2019b). Owing to its location in orographic interior
region, hillslopes receives very low annual rainfall (Fig. 11) and thus comprises least
vegetation on hillslope. The lack of vegetation on hillslopes has been observed to result in



low shear strength of material and hence in instability (Kokutse et al. 2016). Thus,
lithological contrast, rockmass shearing and lack of vegetation are the main reasons of
instability of S.N. 5 landslide. The second landslide (S.N. 19) belongs to inter-layered
schist/gneiss rockmass of the Lesser Himalaya Crystalline (LHC) and is situated at
orographic front where rainfall increases suddenly (Fig. 11). Further, this region is also
subjected to high exhumation rate of 2.0-4.5 mm/yr (Thiede et al. 2009). Therefore,
lithological contrast, high rainfall and high exhumation rate are considered as the main
reasons of instability of this landslide slope.
The landslides that could not result into river damming on the basis of volume and valley
characteristics are mostly in the LHC and Lesser Himalaya Sequence (LHS) region. These
regions consist of a majority of rock fall and rock avalanche type landslides that are generally
of meta-stable ($1 \leq FS \leq 2$) category (Fig. 4). Despite the deep/narrow valley setting, landslides
in these regions may not form the potential landslide dam, at present, owing to relatively less
landslide volume. The possible causes of meta-stability ($1 \leq FS \leq 2$) of rock fall and rock
avalanche may be strong lithology (gneissic), dense vegetation on the hillslopes (Chawla et
al. 2012), relatively less sheared rock mass in comparison to the HHC region, and relatively
less decrease in land use/landcover (Lata et al. 2015). Maximum Total Displacement (TD) is
also associated with the rock fall and rock avalanche of this region (Fig. 4).
In the parametric study, soil friction and in-situ stress are noted to affect the FS most in case
of debris slide, whereas the FS of rock fall and rock avalanche are mainly controlled by 'mi'
i.e., (a Generalized Hoek-Brown criteria parameter) and the in-situ stress. For the TD of the
debris slides, field stress, elastic modulus and Poisson's ratio, whereas for rock falls and rock
avalanches, 'mi' parameter and in-situ stress play the dominant role (Fig. 6,7). Soil friction
(φ) has been a controlling factor for the shear strength and its decrease has been observed to
result in the shear failure of slope material (Matsui and San 1992). The '$m_i$' (a GHB
parameter), an equivalent of the angle of friction of the M-C envelope is observed to
dominate the FS and TD of the rock fall. Since the rainfall constitutes an important role in
decreasing the friction of slope material through percolation and change the pore water
pressure regime (Rahardjo et al. 2005), a relatively high frequency of extreme rainfall events
in the Satluj River valley since the year 2013 (Kumar et al. 2019a) amplifies the risk of
hillslope instability. Furthermore, the in-situ field stress that has been compressional and/or
extensional owing to orogenic setting in the region may also enhance hillslope instability



(Eberhardt et al. 2004; Vannay et al. 2004). Deformation parameters e.g. elastic modulus and
Poisson's ratio are also observed to affect the displacement in slope models of the debris
slides. Similar studies in other regions have also noted the sensitivity of the elastic modulus
and Poisson's ratio on the slope stability (Zhang and Chen 2006).
The study area has been subjected to frequent excessive rainfalls since the year 2010 and
received widespread slope failures and flash-floods (Fig. 11b). Three (S.N. 7,14,15 in Fig. 9)
out of five potential landslide dams that are predicted belong to the Higher Himalaya
Crystalline (HHC) that receives relatively more rainfall (Fig. 11). Previous studies have also
noted that the most of the landslide dams in the river valley had originated in the HHC region
and climatic factors, particularly rainfall, was the most probable reason for the slope failures
(Sharma et al. 2017). The earthquake, however, has been second to rainfall as the triggering
factor for slope failures in the study area. Contrary to the along 'Himalayan' arc distribution
of earthquakes, the study area has received most of the earthquakes around the Kaurik-
Chango Fault only (Fig. 12a). However, the only major earthquake event has been M 6.8
earthquake on 19[th] Jan. 1975 that resulted in widespread landslides (Khattri et al. 1978).
About ~99.5 % of the earthquake events that occurred during the years 1940-2019 in and
around study area had their magnitude less than 6.0 and about ~95 % of all events originated
within 40 km. Such low magnitude seismicity has been attributed to the northward extension
of the Delhi-Haridwar ridge (Hazarika et al. 2019) whereas, shallow seismicity is subjected to
the MHT ramp structure in the region that allows strain accumulation at shallow depth
(Bilham 2019). Thus, earthquake has not been a major landslide triggering process in the
region.
In view of the possible uncertainties in the predictive nature of study, following assumptions
and then resolutions were made;
• To account the effect the spatial variability in slope geometry in the FEM analysis, 3D
models have been in use for the last decade (Griffiths and Marquez 2007). However,
the pre-requisite for the 3D FEM involves the detailed understanding of slope
geometry and material variability in the subsurface that was not possible in the study
area considering steep and inaccessible slopes. Therefore, multiple 2D sections were
chosen, wherever possible. To account the effect of sampling bias and material
variability, a range of values of input parameters was used (sec. 3.4).



- Determination of the debris thickness has been a major problem in landslide volume measurement particularly in steep, narrow river valleys of the NW Himalaya where landslide scarps are not accessible. Therefore, the thickness was approximated by considering the relative altitude of the ground on either side of the deposit, as also performed by Innes (1983). It was assumed that the ground beneath the deposit is regular.

- The resultant dam volume could also be different from the landslide volume due to entrainment of slope material during movement, rockmass fragmentation, pore water pressure, size of debris particles, and washout of landslide material by the river (Hungr and Evans 2004; Dong et al. 2011; Yu et al. 2014). Therefore, dam volume is presumed to be equal to landslide volume for the worst case scenario and least associated uncertainty (sec. 3.5).

- Stream power that is manifested by upstream catchment area and local slope gradient may also vary at temporal scale owing to temporally varying water influx from glaciers and precipitation systems i.e., Indian Summer Monsoon and Western Disturbance (Gadgil et al. 2007; Hunt et al. 2018). Though our study is confined to spatial scale at present, the findings remain subjected to the change at temporal scale.

- The Voellmy rheology based RAMMS model (Voellmy 1955; Salm 1993; Christen et al. 2010) requires calibrated friction ($\mu$) and turbulence ($\xi$) values for the run-out analysis. Though the previous run-out events don't have trace in the study area owing to convergence of landslide toe with the river channel, a range of $\mu$ and $\xi$ values were used in the study in view of material type and run-out path.

Despite these uncertainties, such studies are required to minimize the risk and avert the possible disasters in terrains where human population is bound to live in the proximity of unstable landslides.

**CONCLUSIONS**

Out of forty four landslides that are studied, five landslides are observed to form potential landslide dam, if failure occurs. Though the blocking duration is difficult to predict, upstream and downstream consequences of these damming events can't be overlooked as the region has witnessed many damming and flash floods in the past that resulted in widespread loss of lives and economy.



These five landslides comprise a total landslide volume of 26.3± 6.7 M m$^3$. The slopes of
four landslides (debris slides) out of these five are unstable i.e., Factor of safety <1 whereas,
remaining one (rock avalanche) is meta-stable i.e., $1 \leq FS \geq 2$. Field observations and previous
studies have noted the damming events by these landslides (or the region consisting these
landslides) in the past too. Since the area is witnessing enhanced rainfall and flash floods
since year 2010, findings of the run-out analysis that involve $24.8 \pm 2.7$m to $39.8 \pm 4.0$m high
material flow from these landslides become more crucial.
In order to evaluate the sensitivity of factor of safety and total displacement in slope stability
analysis, parametric study was performed. The angle of internal friction of soil or '$m_i$' (a
parameter of the Generalized Hoek Brown criteria that is equivalent to the angle of internal
friction) of rockmass and *in-situ* field stress are noted to be the most controlling parameters
for the stability of slopes.
**Conflict of Interest**
The authors declare that they have no conflict of interest.
**Dataset Availability**
The dataset (Joints and value of input parameters used in the FEM analysis) is deposited in
approved open access repository (*Mendeley data*) as Kumar et al. (2020).
**Author contribution**
VK collected the field data. VK and IJ performed the laboratory analysis. All authors
contributed to the dataset compilation, numerical modeling (Slope stability and Run-out) and
geomorphic interpretations. All authors contributed to the writing of final draft.
**ACKNOWLEDGEMENT**
The authors are thankful to the Director, Wadia Institute of Himalayan Geology (WIHG) for
all the necessary support. VK and IJ acknowledge constructive discussion on the regional
scale study with Prof. H.B. Havenith, Prof. D.V. Griffiths, and Prof. D.P. Kanungo. VG and
RKB acknowledge the financial help through project MOES/Indo-Nor/PS-2/2015. Authors
are thankful to the RAMMS developer for the license.

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

**LIST OF FIGURES AND TABLES**
**Fig. 1**   Geological setting. TS, HHC, LHC and LHS are Tethyan Sequence, Higher Himalaya

Crystalline, Lesser Himalaya Crystalline and Lesser Himalaya Sequence, respectively.

WGC: Wangtu Gneissic Complex. Geological setting is based on Sharma (1977);

Vannay et al., (2001); Kumar et al., (2019b). The red dashed circle in the inset

represents the region within 100 km radius from the Satluj River (marked as blue line)

that was used to determine the earthquake distribution in the area. KCF in inset refers to

Kaurik-Chango Fault. The numbers 1-44 refer to serial number of landslides.

**Fig. 2**   Field photographs of some of the landslides (a) Khokpa landslide (**S.N.1**); (b)

Akpa_III landslide (**S.N. 5**); (c) Rarang landslide (**S.N. 6**); (d) Pawari landslide (**S.N.14**);

(e) Urni landslide (**S.N.19**); (f) Barauni Gad_I_S landslide(**S.N. 38**).Black circle in the

pictures that encircles the vehicle is intended to represent the relative scale.

**Fig. 3**   TheFEM configuration ofsome of the slope models. S.N. refers to serial no. of

landslides in Table 1. The joint distribution model in all the slopes was parallel statistical

with normal distribution of joint spacing.

**Fig. 4**   The FEM analysis of all forty-four landslides. Grey bar in the background highlights

the Higher Himalaya Crystalline (HHC) region that comprises relatively more unstable

landslides, relatively more landslide volume and human population. Source of human

population: Census 2011 (Govt. of India, New Delhi).TS, KKG, HHC, LHC and LHS

are Tethyan Sequence, Kinnaur Kailash Granite, Higher Himalaya Crystalline, Lesser

Himalaya Crystalline and Lesser Himalaya Sequence, respectively



**Fig. 5** Relationship of Factor of Safety (FS), Total Displacement (TD) and Shear Strain (SS). DS, RF, and RA refer to Debris slide, rock fall and rock avalanche, respectively.

**Fig. 6** Parametric analysis of debris slides. (a) Akpa_III (S.N. 5): (b) Pangi_III (S.N. 13): (c) Barauni Gad_I_S(S.N. 38). S. N. refers to serial no. of landslides in Table 1.

**Fig. 7** Parametric analysis of rockfall/rock avalanche. (a) Tirung khad (S.N. 2): (b) Baren Dogri (S.No. 7): (c) Chagaon_II (S.N. 21).

**Fig. 8** Landslide damming indices (a) Morphological Obstruction Index (MOI); (b) Hydro-morphological dam stability index (HDSI); (c) Landslides vs. MOI; (d) Landslides vs. HDSI.

**Fig. 9** Potential landslide damming locations. (a) Akpa_III landslide; (b) Baren dogri landslide; (c) Pawari landslide; (d) Telangi landslide; (e) Urni landslide. S. N. refers to serial no. of landslides in Table 1.

**Fig. 10** Field signatures of the landslide damming near Akpa_IIIlandslide. (a) Upstream view of Akpa landslide with lacustrine deposit at the left bank; (b) enlarged view of lacustrine deposit with arrow indicating lacustrine sequence; (c) alternating fine-coarse sediments. F and Crefer to fine (covered by yellow dashed lines) and coarse (covered by green dashed lines) sediments, respectively.

**Fig. 11** Rainfall distribution. (a) Topographic profile; (b) annual rainfall; (c) monsoonal (June-Sep.) rainfall; (d) non-monsoonal (Oct.-May) rainfall. Green bars represent years of relatively more rainfall resulting into flash floods, landslides and socio-economic loss in the region. (i):hpenvis.nic.in, retrieved on March 1, 2020; Department of Revenue, Govt. of H.P. (ii): hpenvis.nic.in, retrieved on March 1, 2020.(iii): Kumar et al., 2019a;ndma.gov.in, retrieved on march 1, 2020 (iv):sandrp.in, retrieved on march 1, 2020.The numbers 1-44 refer to serial number of landslides.

**Fig. 12** Earthquake distribution. (a) Spatial variation of earthquakes. The transparent circle represents the region within 100 km radius from the Satluj River (blue line). The black dashed line represents the seismic dominance around Kaurik-Chango fault;(b) earthquake magnitude vs. focal depth. The red dashed region highlights the concentration of earthquakes within 40 km depth. ISC: International Seismological





Centre; (c) Cross section view (Based on Hazarika et al. 2017; Bilham, 2019). Red
dashed circle represents the zone of strain accumulation caused by the Indian and
Eurasian plate collision. SD, MCT, MT, MBT and HFT are Sangla Detachment, Main
Central Thrust, Munsiari Thrust, Main Boundary Thrust and Himalayan Frontal Thrust,
respectively.
**Fig. 13** Results of run-out analysis. μ refers to coefficient of friction. S. N. refers to serial no.
of landslides in Table 1
**Fig. 14** Results of run-out analysis at different values of μ and ξ. μ and ξ refer to coefficient
of friction and turbulence, respectively.
**Table 1** Details of landslides used in the study.
**Table 2** Details of satellite imagery.
**Table 3** Criteria used in the Finite Element Method (FEM) analysis.
**Table 4** Details of input parameters used in run-out analysis.



| S.N. | Landslide location | Latitude/ Longitude | Type | Area[1], m$^2$ | Volume[2], m$^3$ | Human population[3] | Litho-tectonic division |
|---|---|---|---|---|---|---|---|
| 1 | Khokpa | 31°35'18.9"N 78°26'28.6"E | Debris slide | 21897± 241 | 43794± 18361 | 373 | Tethyan Sequence (TS) |
| 2 | Tirung Khad | 31°34'50.4"N 78°26'20.5"E | Rockfall | 28537± 314 | 14269± 9055 | 0 | |
| 3 | Akpa _I | 31°34'57.1"N 78°24'30.6"E | Rock avalanche | 963051± 10594 | 1926102± 807515 | 0 | TS-KKG |
| 4 | Akpa_II | 31°35'2.2"N 78°23'25.4"E | Rock avalanche | 95902± 1055 | 143853± 40734 | 470 | Kinnaur Kailash Granite (KKG) |
| 5 | Akpa_III | 31°34'54.5"N 78°23'2.4"E | Debris slide | 379570± 4175 | 7591400± 3182681 | 1617 | |
| 6 | Rarang | 31°35'58.7"N 78°20'39.1"E | Rockfall | 4586± 50 | 4586± 1923 | 848 | |
| 7 | Baren Dogri | 31°36'23.6"N 78°20'23.1"E | Rock avalanche | 483721± 5321 | 2418605±421561 | 142 | |
| 8 | Thopan Dogri | 31°36'12.3"N 78°19'50.4"E | Rockfall | 55296± 608 | 165888± 46974 | 103 | |
| 9 | Kashang Khad_I | 31°36'5.0"N 78°18'44.4"E | Debris slide | 113054± 1244 | 169581± 48019 | 103 | |
| 10 | Kashang Khad_II | 31°35'58.3"N 78°18'34.0"E | Rockfall | 27171± 299 | 40757± 11541 | 103 | |
| 11 | Pangi _I | 31°35'36.4"N 78°17'36.4"E | Debris slide | 30112± 331 | 45168± 12790 | 1389 | Higher Himalaya Crystalline (HHC) |
| 12 | Pangi _II | 31°35'38.9"N 78°17'12.2"E | Debris slide | 59436± 654 | 118872± 49837 | 1389 | |
| 13 | Pangi _III | 31°34'38.9"N 78°16'55.6"E | Debris slide | 75396± 829 | 188490± 32854 | 7 | |
| 14 | Pawari | 31°33'49.8"N 78°16'28.6"E | Debris slide | 320564± 3526 | 1602820± 279370 | 4427 | |
| 15 | Telangi | 31°33'7.0"N 78°16'37.2"E | Debris slide | 543343± 5977 | 13583575± 2367608 | 6817 | |
| 16 | Shongthong | 31°31'13.0"N 78°16'17.0"E | Debris slide | 5727± 63 | 11454± 2464 | 388 | |
| 17 | Karchham | 31°30'12.4"N 78°11'30.8"E | Rock avalanche | 28046± 309 | 56092± 23516 | 0 | |
| 18 | Choling | 31°31'17.0"N 78° 8'4.9"E | Debris slide | 20977± 231 | 20977± 8795 | 0 | Lesser Himalaya Crystalline (LHC) |
| 19 | Urni | 31°31'8.0"N 78° 7'42.2"E | Debris slide | 112097± 1233 | 1120970± 469965 | 500 | |
| 20 | Chagaon_I | 31°30'55.9"N 78° 6'52.0"E | Rockfall | 3220± 35 | 3220± 1350 | 0 | |
| 21 | Chagaon_II | 31°30'57.9"N 78° 6'47.7"E | Rockfall | 11652± 128 | 11652± 4885 | 0 | |



| | | | | | | |
|---|---|---|---|---|---|---|
| 22 | Chagaon_III | 31°31'3.0"N 78° 6'21.4"E | Debris slide | 42141± 464 | 168564± 70670 | 1085 | |
| 23 | Wangtu_U/s | 31°32'4.8"N 78° 3'5.0"E | Rock avalanche | 211599± 2328 | 317399± 89876 | 17 | |
| 24 | Wangtu D/s__1 | 31°33'27.7"N 77°59'43.7"E | Debris slide | 4655± 51 | 9310± 3903 | 71 | |
| 25 | Kandar | 31°33'43.7"N 77°59'54.9"E | Rock avalanche | 151128± 1662 | 302256± 126720 | 186 | |
| 26 | Wangtu D/s_ 2 | 31°33'38.9"N 77°59'29.9"E | Debris slide | 8004± 88 | 16008± 6711 | 71 | |
| 27 | Agade | 31°33'52.3"N 77°58'3.5"E | Debris slide | 9767± 107 | 14651± 4149 | 356 | |
| 28 | Punaspa | 31°33'37.6"N 77°57'31.5"E | Debris slide | 3211± 35 | 3211± 1346 | 343 | |
| 29 | Sungra | 31°33'58.8"N 77°56'49.6"E | Debris slide | 5560± 61 | 11120± 4662 | 2669 | |
| 30 | Chota Kamba | 31°33'39.2"N 77°54'39.0"E | Rock avalanche | 197290± 2170 | 591870± 167597 | 401 | |
| 31 | Bara Kamba | 31°34'10.4"N 77°52'56.7"E | Rockfall | 36347± 400 | 18174± 7619 | 564 | |
| 32 | Karape | 31°33'44.9"N 77°53'13.9"E | Debris slide | 50979± 561 | 50979± 21373 | 1118 | |
| 33 | Pashpa | 31°34'40.2"N 77°50'53.0"E | Rockfall | 16079± 171 | 8040± 3371 | 29 | |
| 34 | Khani Dhar_I | 31°33'43.4"N 77°48'52.5"E | Rock avalanche | 218688± 2406 | 874752± 366738 | 0 | |
| 35 | Khani Dhar_II | 31°33'26.3"N 77°48'35.8"E | Rock avalanche | 146994± 1617 | 734970± 248125 | 0 | |
| 36 | Khani Dhar_III | 31°33'20.1"N 77°48'27.8"E | Rock avalanche | 20902± 230 | 62706± 17756 | 0 | |
| 37 | Jeori | 31°31'58.8"N 77°46'18.2"E | Rock avalanche | 93705± 1031 | 93705± 39286 | 0 | |
| 38 | Barauni Gad_I_S | 31°28'56.6"N 77°41'40.4"E | Debris slide | 63241± 696 | 758892± 111620 | 236 | LHC-LHS |
| 39 | Barauni Gad_I_Q | 31°29'00.0"N 77°41'38.0"E | Debris slide | 59273± 652 | 711276± 104616 | 0 | Lesser Himalaya Sequence (LHS) |
| 40 | Barauni Gad_II | 31°28'43.9"N 77°41'24.6"E | Rockfall | 6977± 77 | 3489± 1463 | 0 | |
| 41 | Barauni Gad_III | 31°29'5.6"N 77°41'23.7"E | Rockfall | 33115± 364 | 33115± 13883 | 0 | |
| 42 | D/s Barauni Gad_I | 31°28'24.9"N 77°41'8.4"E | Rockfall | 19101± 210 | 19101± 8008 | 0 | |
| 43 | D/s Barauni Gad_II | 31°28'25.5"N 77°40'56.7"E | Rockfall | 21236± 234 | 21236± 8903 | 0 | |
| 44 | D/s Barauni Gad_III | 31°28'7.4"N 77°40'42.4"E | Rockfall | 15632± 172 | 15632± 6554 | 0 | |





[1]Error (±) caused by GE measurement (1.06 %).

[2]Error (±) is an outcome of multiplication of area ± error and thickness ± error. Thickness error (Std. dev.) corresponds to averaging of field based approximated thickness.

[3]The human population is based on census 2011, Govt. of India. The villages/town in the radius of 500 m from the landslide are considered to count the human population.

**Table 1**   Details of landslides used in the study.





| Satellite data | | Source | Date of data | Spatial resolution |
|---|---|---|---|---|
| CARTOSAT-1 stereo imagery | 524/253 | National Remote Sensing Center (NRSC), Hyderabad, India | 5th Dec. 2010 | ~2.5 m |
| | 525/253 | | 16th Dec. 2010 | ~2.5 m |
| | 526/252 | | 18th Oct. 2011 | ~2.5 m |
| | 526/253 | | 18th Oct. 2011 | ~2.5 m |
| | 527/252 | | 24th Nov. 2010 | ~2.5 m |
| | 527/253 | | 27th Dec. 2010 | ~2.5 m |
| | 528/252 | | 26th Nov. 2011 | ~2.5 m |

**Table 2** Details of satellite imagery.





**Table 3** Criteria used in the Finite Element Method (FEM) analysis.

| Material Criteria | Parameters | Source |
|---|---|---|
| **Generalized Hoek & Brown (GHB) Criteria** (Hoek et al. 1995)<br><br>$\sigma_1 = \sigma_3 + \sigma_{ci}[m_b(\sigma_3/\sigma_{ci}) + s]^a$<br><br>Here, $\sigma_1$ and $\sigma_3$ are major and minor effective principal stresses at failure; $\sigma_{ci}$, compressive strength of intact rock; $m_b$, a reduced value of the material constant ($m_i$) and is given by;<br><br>$m_b = m_i e^{[(GSI-100)/(28-14D)]}$<br><br>s and a; constants for the rock mass given by the following relationships;<br><br>$s = e^{[(GSI-100)/(9-3D)]}$<br>$a = \frac{1}{2} + \frac{1}{6}\left[e^{[-(\frac{GSI}{15})]} - e^{[-(\frac{20}{3})]}\right]$<br><br>Here, D; a factor which depends upon the degree of disturbance to which the rock mass has been subjected by blast damage and stress relaxation. GSI (Geological Strength Index); a rockmass characterization parameter. | Unit Weight, $\gamma$ (MN/m$^3$) | Laboratory analysis (UCS) (IS: 9143-1979) |
| | Uniaxial Compressive Strength, $\sigma_{ci}$ (MPa) | |
| | Rockmass modulus (MPa) | Laboratory analysis (Ultrasonic velocity test); Hoek and Diederichs (2006). |
| | Poisson's Ratio | |
| | Geological Strength Index | Field observation and based on recent amendments (Cai et al. 2007 and reference therein) |
| | Material Constant ($m_i$) | Standard values (Hoek and Brown 1997) |
| | $m_b$ | GSI was field dependent, $m_i$ as per (Hoek and Brown 1997) and D is used between 0-1 in view of rockmass exposure and blasting. |
| | s | |
| | a | |
| | D | |
| **Barton-Bandis Criteria** (Barton and Choubey 1977; Barton and Bandis 1990)<br><br>$\tau = \sigma_n \tan[\emptyset_r + JRC \log_{10}(JCS/\sigma_n)]$<br><br>Here, $\tau$ is joint shear strength; $\sigma_n$, normal stress across joint; $\emptyset_r$, reduced friction angle; JRC, joint roughness coefficient; JCS, joint compressive strength.<br><br>JRC is based on the chart of Barton and Choubey (1977); Jang et al. (2014). JCS was determined using following equation;<br><br>$\log_{10}(JCS) = 0.00088 (R_L)(\gamma) + 1.01$<br><br>Here, $R_L$ isSchimdt Hammer Rebound value and $\gamma$ is unit weight of rock.<br><br>The JRC and JCS were used as $JRC_n$ and $JCS_n$. following the scale corrections observed by Barton and | Normal Stiffness, $k_n$ (MPa/m) | $E_i$ is lab dependent.L and GSI were field depenedent. D is used between 0-1 in view of rockmass exposure and blasting. |
| | Shear Stiffness, $k_s$ (MPa/m) | It is assumed as $k_n/10$. However, effect of denominator is aslo obtainedthrough parameteric study. |
| | Reduced friction angle, $\emptyset_r$ | Standard values ( Barton and Choubey 1977). |
| | Joint roughness coefficient, JRC | Field based data from profilometer and standard values from Barton and Choubey (1977); Jang et al. (2014). |





| | | Joint compressive strength, JCS (MPa) | Empirical equationof Deere and Miller (1966) relating Schimdt Hammer Rebound (SHR) values, $\sigma_{ci}$ and unit weight of rock. SHR was field dependent. |
|---|---|---|---|
| | Choubey (1977) and reference therein and proposed by Barton and Bandis (1982).<br><br>$JRC_n = [JRC(L/L_o)^{-0.02(JRC)}]$<br>$JCS_n = [JCS(L/L_o)^{-0.03(JRC)}]$<br><br>Here, Land $L_o$ are mean joint spacing in field and, respectively. $L_o$ has been suggested to be 10 cm.<br><br>**Joint stiffness criteria**<br>(Barton 1972)<br><br>$k_n = (E_i \ast E_m)/L \ast (E_i - E_m)$<br>Here, $k_n$; Normal stiffness, $E_i$; Intact rock modulus, $E_m$; Rockmass modulus L; Mean joint spacing.<br><br>$E_m = (E_i) \ast [0.02 + \{1-D/2\}/\{1 + e^{(60+15 \ast D-GSI)/11)}\}]$<br><br>Here, $E_m$ is based on Hoek and Diederichs (2006) and reference therein | Scale corrected, $JRC_n$ | Empirical equation of Barton and Bandis (1982). |
| | | Scale corrected, $JCS_n$ (MPa) | |
| Soil | **Mohr-Coulomb Criteria**<br>(Coulomb 1776; Mohr 1914)<br><br>$\boldsymbol{\tau = C + \sigma \, tan\emptyset}$<br><br>Here, $\tau$; Shear stress at failure, C; Cohesion, $\sigma_n$; normal strength, Ø; angle of friction. | Unit Weight (MN/m$^3$) | Laboratory analysis (UCS) (IS: 2720-Part 4–1985; IS: 2720-Part 10-1991) |
| | | Young's Modulus, $E_i$ (MPa) | Laboratory analysis (UCS); IS: 2720-Part 10-1991. |
| | | Poisson's Ratio | Standard values from Bowles (1996) |
| | | Cohesion, C (MPa) | Laboratory analysis (Direct shear) |
| | | Friction angle, Ø | (IS: 2720-Part 13- 1986) |



| Landslide | Material type | Material depth[1], m | Friction coefficient[2] | Turbulence coefficient[3], m/sec[2] |
|-----------|---------------|----------------------|-------------------------|-------------------------------------|
| Akpa (S.N. 5) | Gravelly sand | 5 | $\mu$= 0.05, 0.1, 0.3 | $\xi$ = 100, 200, 300 |
| Baren Dogri (S.N. 7) | Gravelly sand | 1.25 | $\mu$= 0.05, 0.1, 0.4 | $\xi$ = 100, 200, 300 |
| Pawari (S.N. 14) | Gravelly sand | 1.25 | $\mu$= 0.05, 0.1, 0.4 | $\xi$ = 100, 200, 300 |
| Telangi (S.N. 15) | Gravelly sand | 6.25 | $\mu$= 0.05, 0.1, 0.4 | $\xi$ = 100, 200, 300 |
| Urni (S.N. 19) | Gravelly sand | 2.5 | $\mu$= 0.06, 0.1, 0.4 | $\xi$ = 100, 200, 300 |

[1] Considering that fact that during slope failure, irrespective of type of trigger, entire loose material might not slide down, the depth is taken as only ¼ (thickness) in the calculation. [2] Since the angle of run-out track (slope and river channel) varied a little beyond the suggested range 2.8º -21.8º or $\mu$ = 0.05-0.4 (Hungr et al., 1984; RAMMS v.1.7.0), we kept out input in this suggested range wherever possible to avoid simulation uncertainty. [3] This range is used in view of the type of loose material i.e., granular in this study (RAMMS v.1.7.0).

**Table 4** Details of input parameters for run-out analysis. S.N. refers to serial number of landslides in Fig. 1.



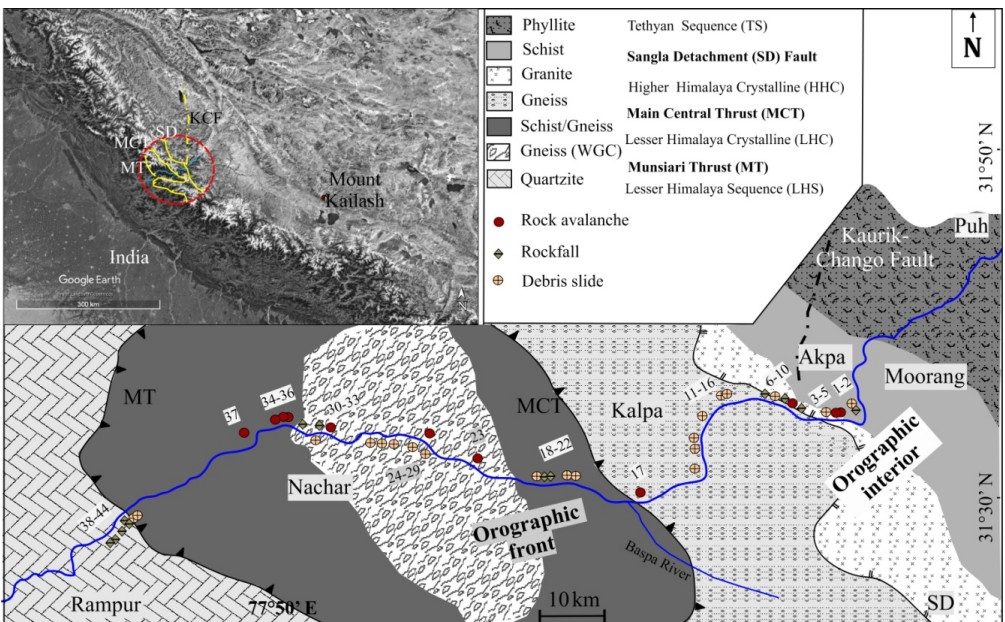

**Fig. 1** Geological setting. TS, HHC, LHC and LHS are Tethyan Sequence, Higher Himalaya Crystalline, Lesser Himalaya Crystalline and Lesser Himalaya Sequence, respectively. WGC: Wangtu Gneissic Complex. Geological setting is based on Sharma (1977); Vannay et al., (2001); Kumar et al., (2019b). The red dashed circle in the inset represents the region within 100 km radius from the Satluj River (marked as blue line) that was used to determine the earthquake distribution in the area. KCF in inset refers to Kaurik-Chango Fault. The numbers 1-44 refer to serial number of landslides.





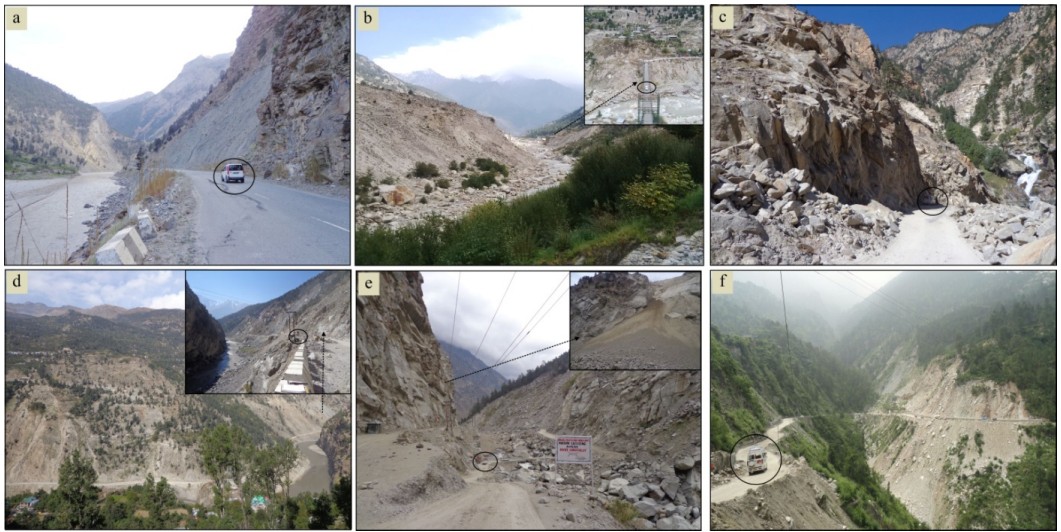

**Fig. 2** Field photographs of some of the landslides (a) Khokpa landslide (**S.N.1**); (b) Akpa_III landslide (**S.N. 5**); (c) Rarang landslide (**S.N. 6**); (d) Pawari landslide (**S.N.14**); (e) Urni landslide (**S.N.19**); (f) Barauni Gad_I_S landslide(**S.N. 38**).Black circle in the pictures that encircles the vehicle is intended to represent the relative scale.



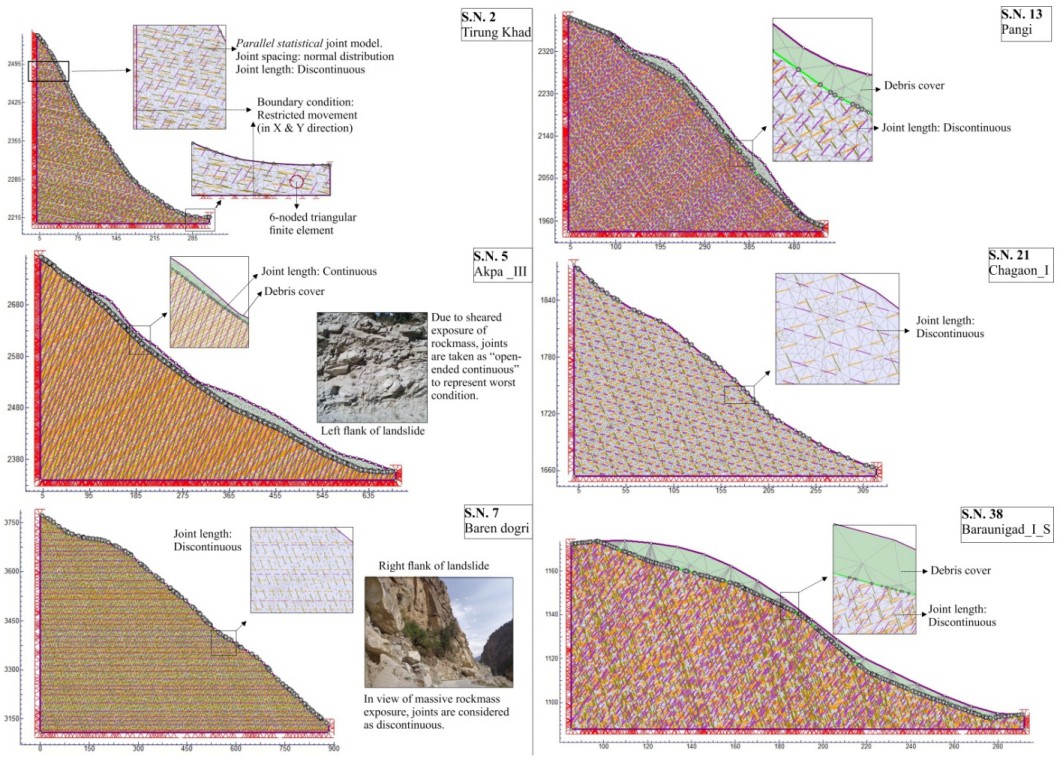

**Fig. 3** TheFEM configuration ofsome of the slope models. S.N. refers to serial no. of landslides in Table 1. The joint distribution model in all the slopes was parallel statistical with normal distribution of joint spacing.



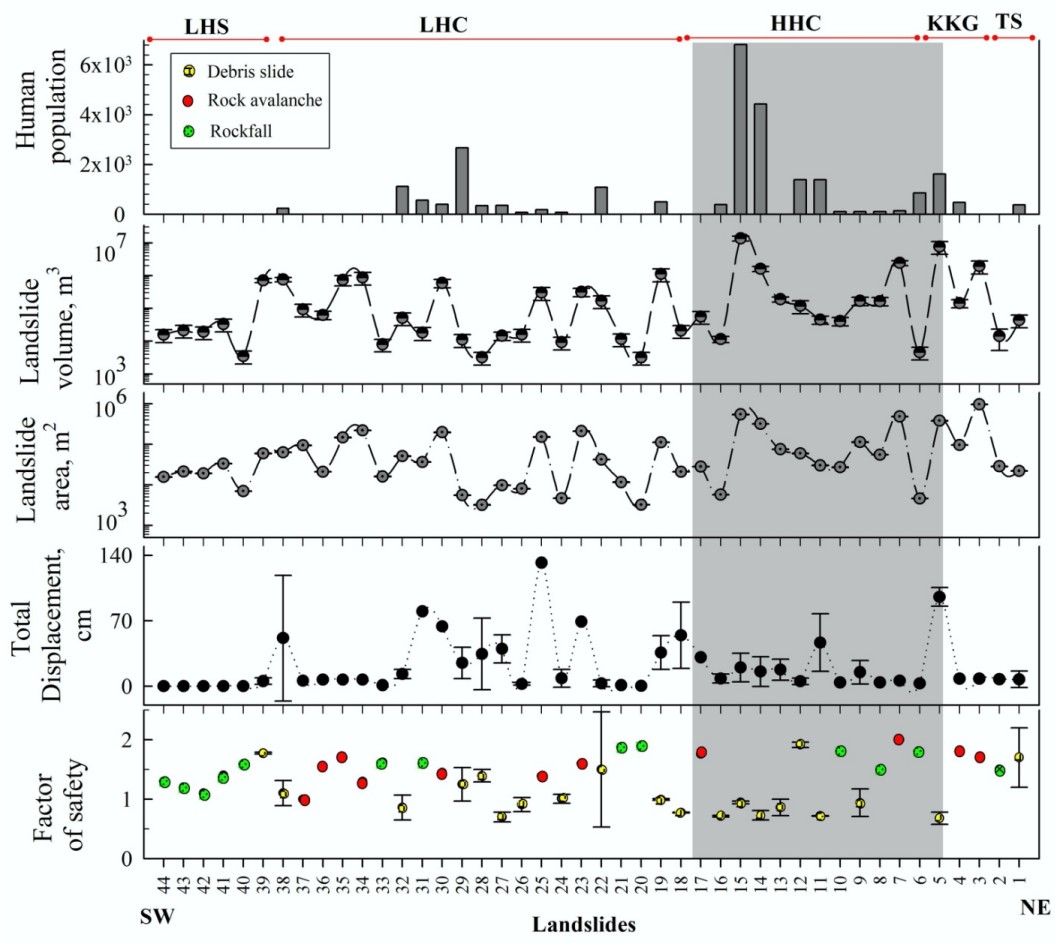

**Fig. 4** The FEM analysis of all forty-four landslides. Grey bar in the background highlights the Higher Himalaya Crystalline (HHC) region that comprises relatively more unstable landslides, relatively more landslide volume and human population. Source of human population: Census 2011 (Govt. of India, New Delhi).TS, KKG, HHC, LHC and LHS are Tethyan Sequence, Kinnaur Kailash Granite, Higher Himalaya Crystalline, Lesser Himalaya Crystalline and Lesser Himalaya Sequence, respectively



Earth **Surface**
Dynamics
Discussions

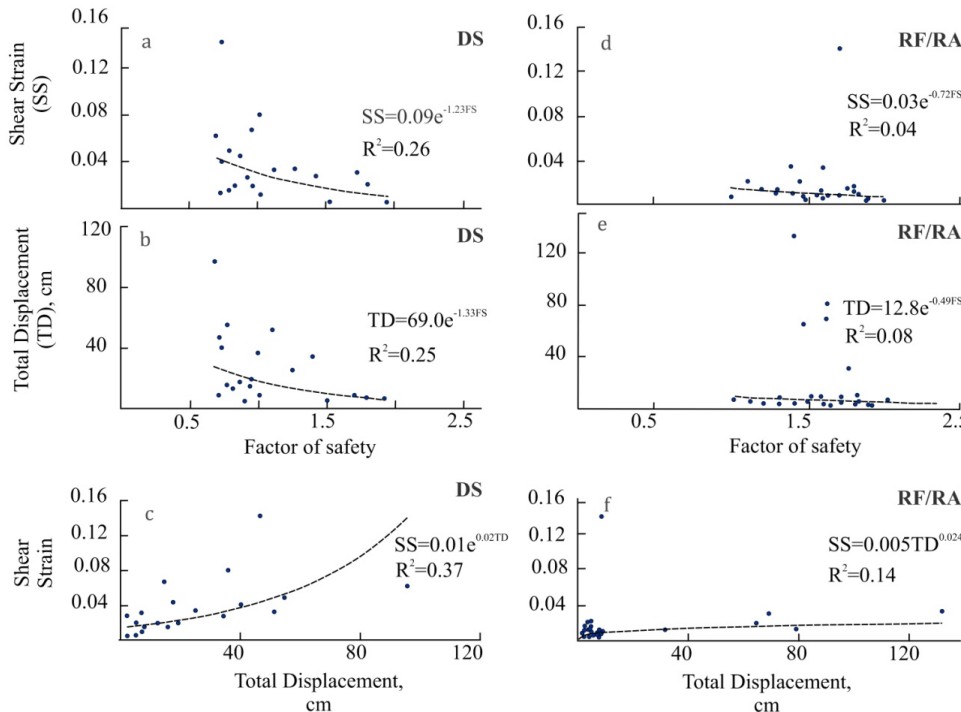

**Fig. 5** Relationship of Factor of Safety (FS), Total Displacement (TD) and Shear Strain (SS). DS, RF, and RA refer to Debris slide, rock fall and rock avalanche, respectively.





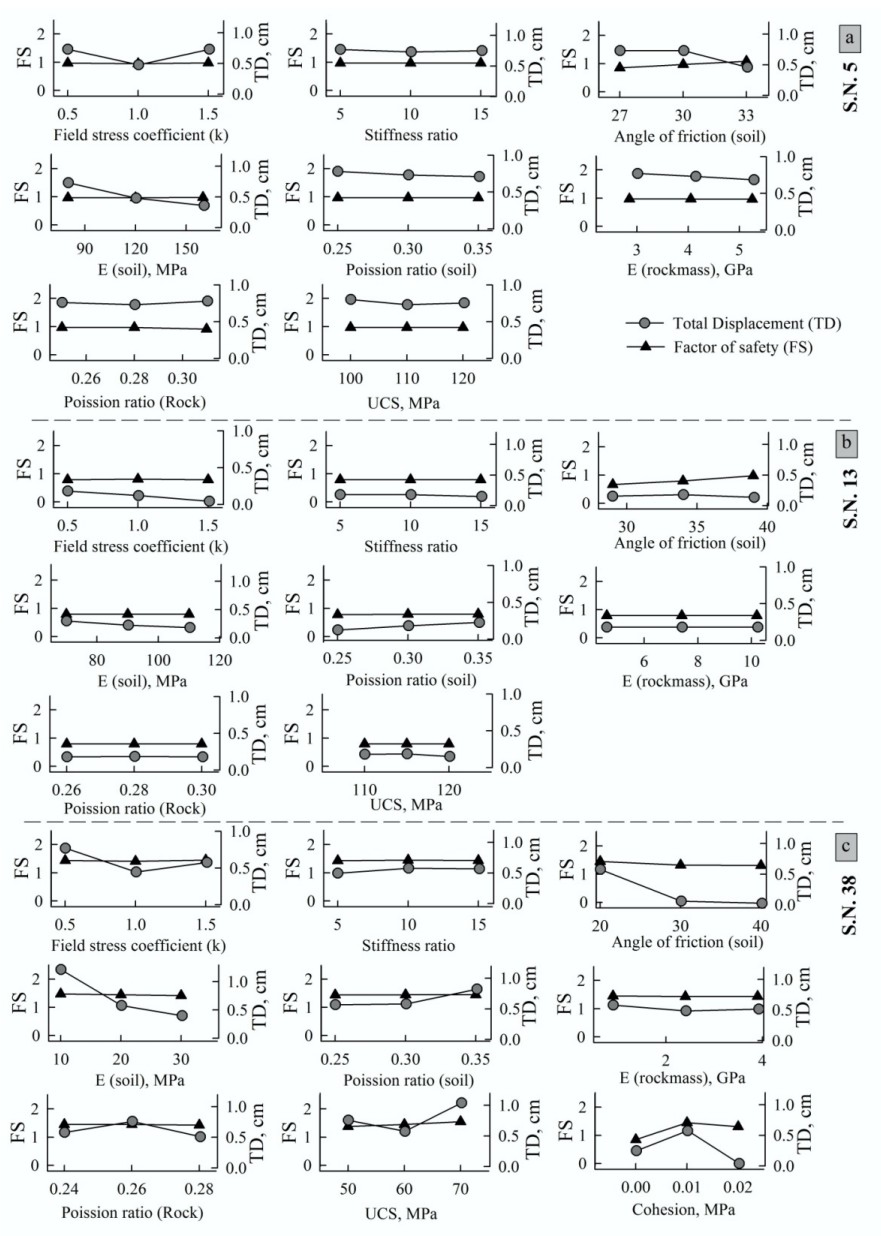

**Fig. 6** Parametric analysis of debris slides. (a) Akpa_III (S.N. 5): (b) Pangi_III (S.N. 13): (c) Barauni Gad_I_S(S.N. 38). S. N. refers to serial no. of landslides in Table 1.



Earth **Surface**
**Dynamics**
Discussions



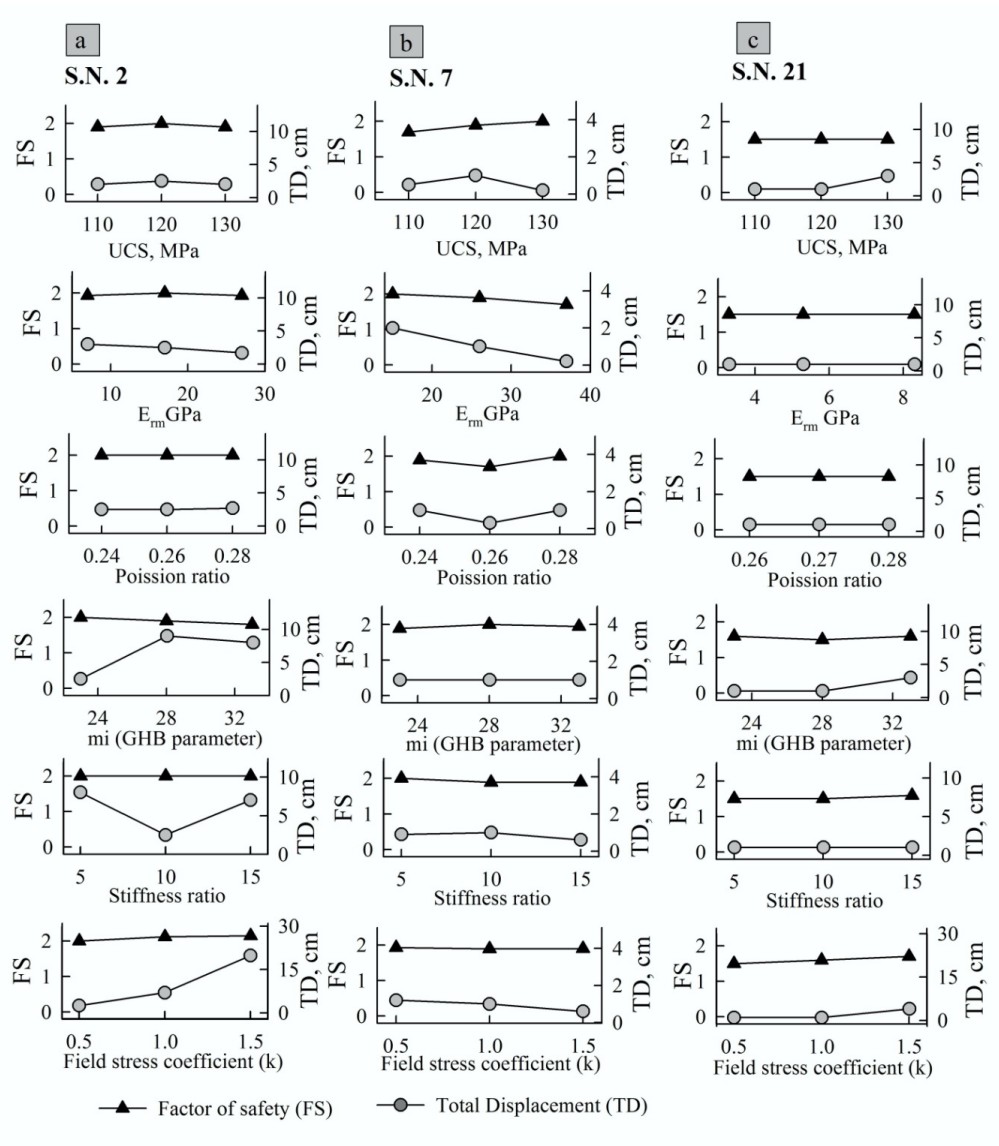

**Fig. 7** Parametric analysis of rockfall/rock avalanche. (a) Tirung khad (S.N. 2): (b) Baren Dogri (S.No. 7): (c) Chagaon_II (S.N. 21).



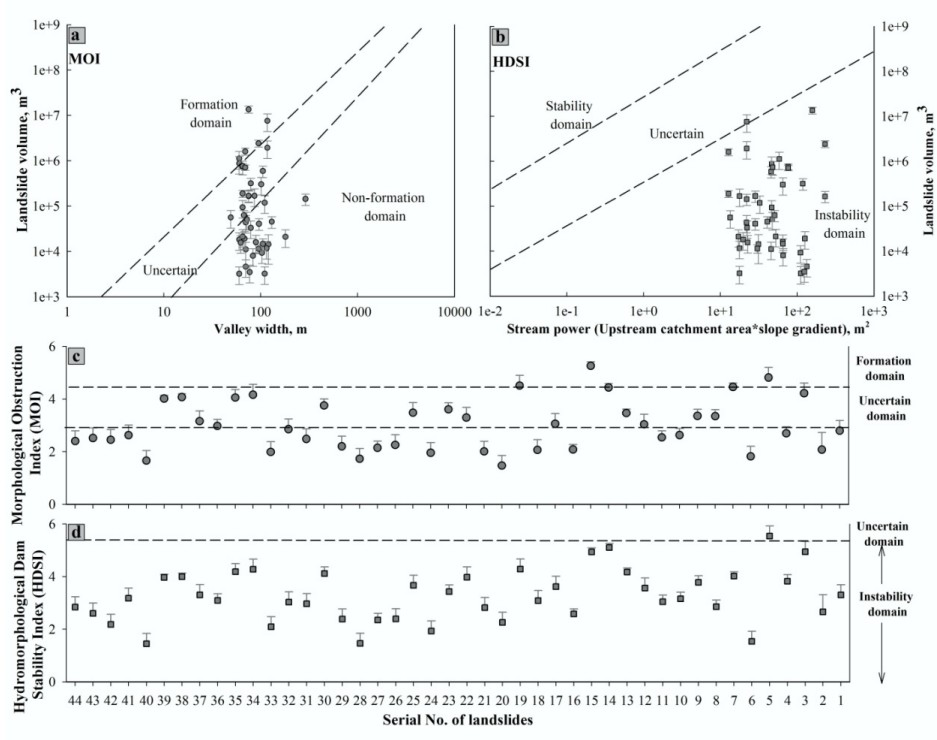

**Fig. 8** Landslide damming indices (a) Morphological Obstruction Index (MOI); (b) Hydro-morphological dam stability index (HDSI); (c) Landslides vs. MOI; (d) Landslides vs. HDSI.



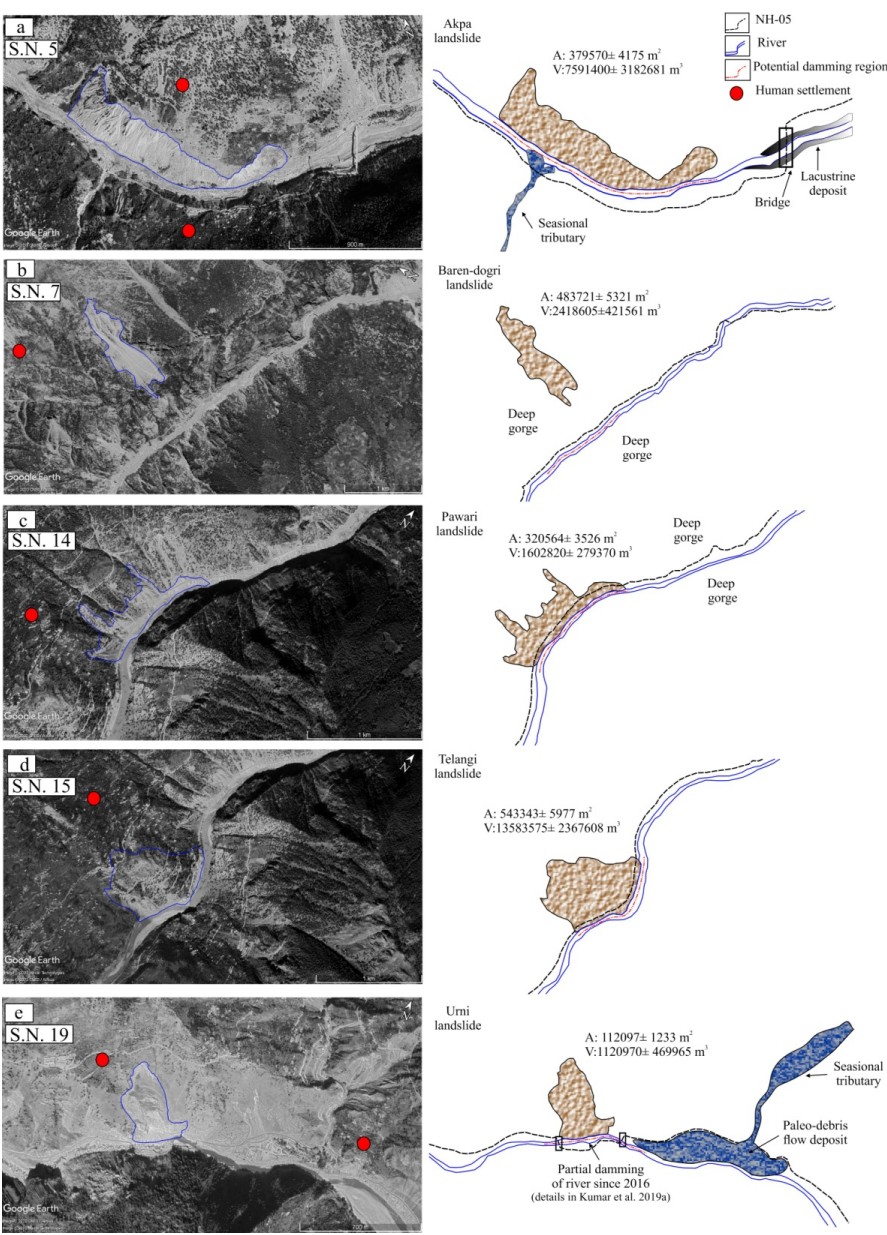

**Fig. 9** Potential landslide damming locations. (a) Akpa_III landslide; (b) Baren dogri landslide; (c) Pawari landslide; (d) Telangi landslide; (e) Urni landslide. S. N. refers to serial no. of landslides in Table 1.



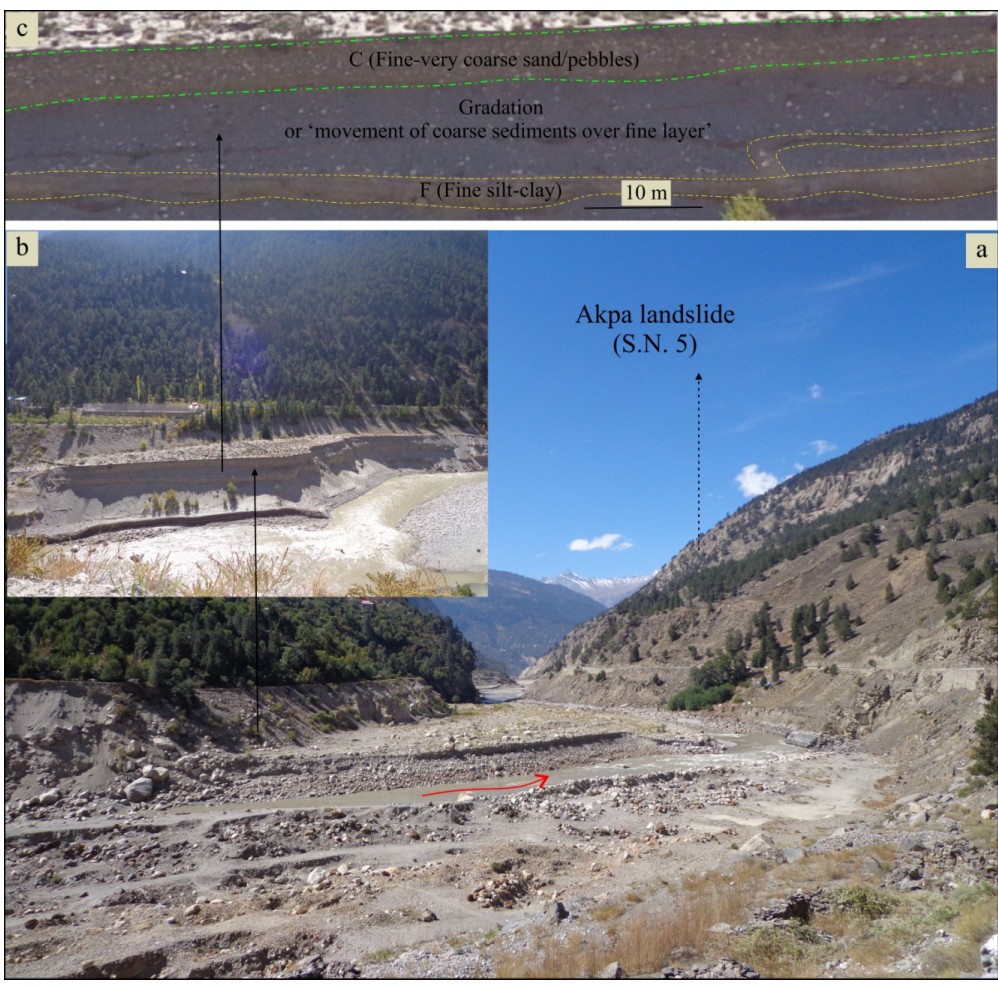

**Fig. 10** Field signatures of the landslide damming near Akpa_IIIlandslide. (a) Upstream view of Akpa landslide with lacustrine deposit at the left bank; (b) enlarged view of lacustrine deposit with arrow indicating lacustrine sequence; (c) alternating fine-coarse sediments. F and Crefer to fine (covered by yellow dashed lines) and coarse (covered by green dashed lines) sediments, respectively.



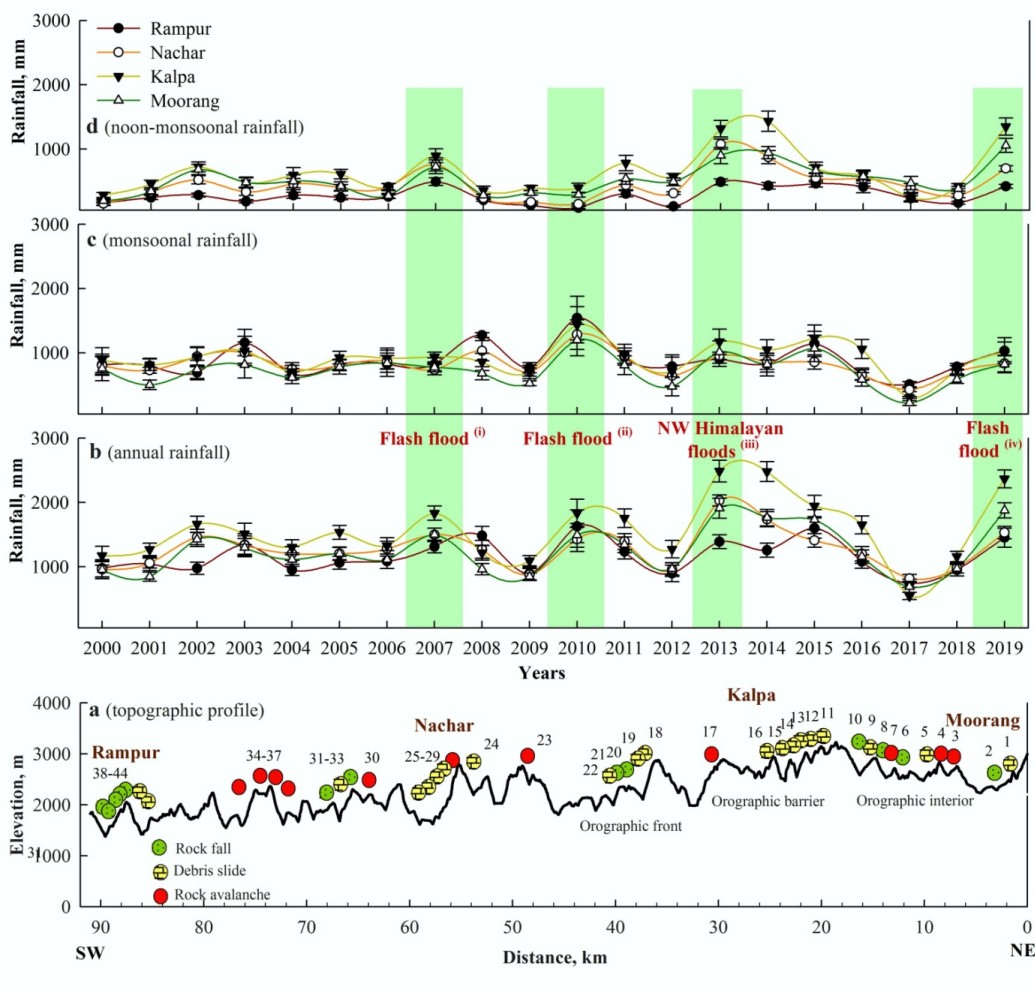

**Fig. 11** Rainfall distribution. (a) Topographic profile; (b) annual rainfall; (c) monsoonal (June-Sep.) rainfall; (d) non-monsoonal (Oct.-May) rainfall. Green bars represent years of relatively more rainfall resulting into flash floods, landslides and socio-economic loss in the region. (i):hpenvis.nic.in, retrieved on March 1, 2020; Department of Revenue, Govt. of H.P. (ii): hpenvis.nic.in, retrieved on March 1, 2020.(iii): Kumar et al., 2019a;ndma.gov.in, retrieved on march 1, 2020 (iv):sandrp.in, retrieved on march 1, 2020.The numbers 1-44 refer to serial number of landslides.



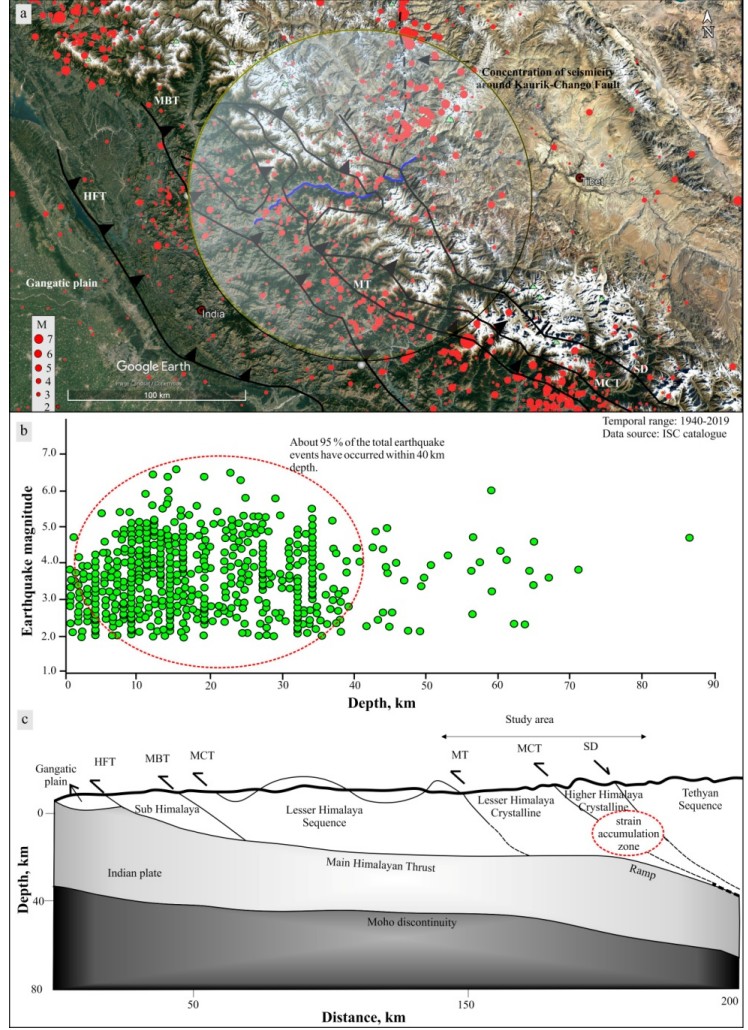

**Fig. 12** Earthquake distribution. (a) Spatial variation of earthquakes. The transparent circle represents the region within 100 km radius from the Satluj River (blue line). The black dashed line represents the seismic dominance around Kaurik-Chango fault;(b) earthquake magnitude vs. focal depth. The red dashed region highlights the concentration of earthquakes within 40 km depth. ISC: International Seismological Centre; (c) Cross section view (Based on Hazarika et al. 2017; Bilham, 2019). Red dashed circle represents the zone of strain accumulation caused by the Indian and Eurasian plate collision. SD, MCT, MT, MBT and HFT are Sangla Detachment, Main Central Thrust, Munsiari Thrust, Main Boundary Thrust and Himalayan Frontal Thrust, respectively.





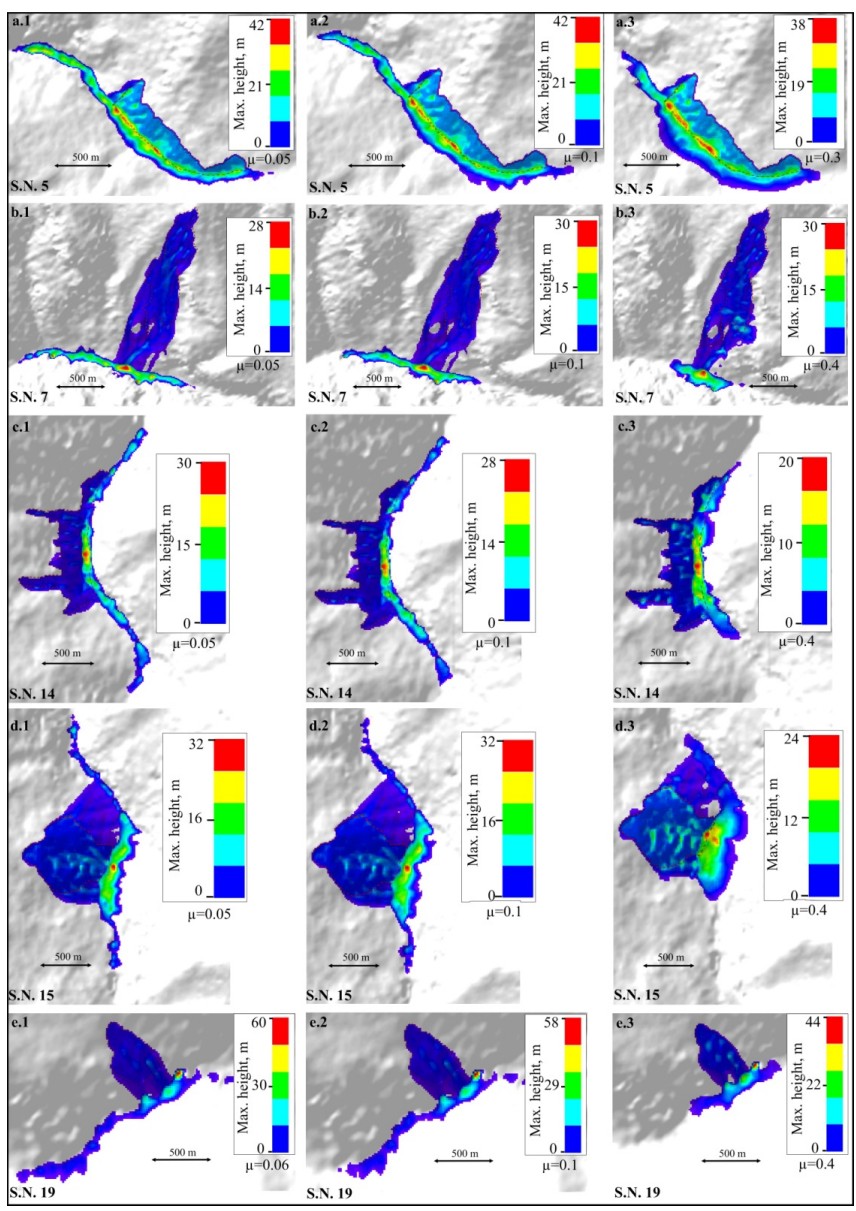

**Fig. 13** Results of run-out analysis. μ refers to coefficient of friction. S. N. refers to serial no. of landslides in Table 1



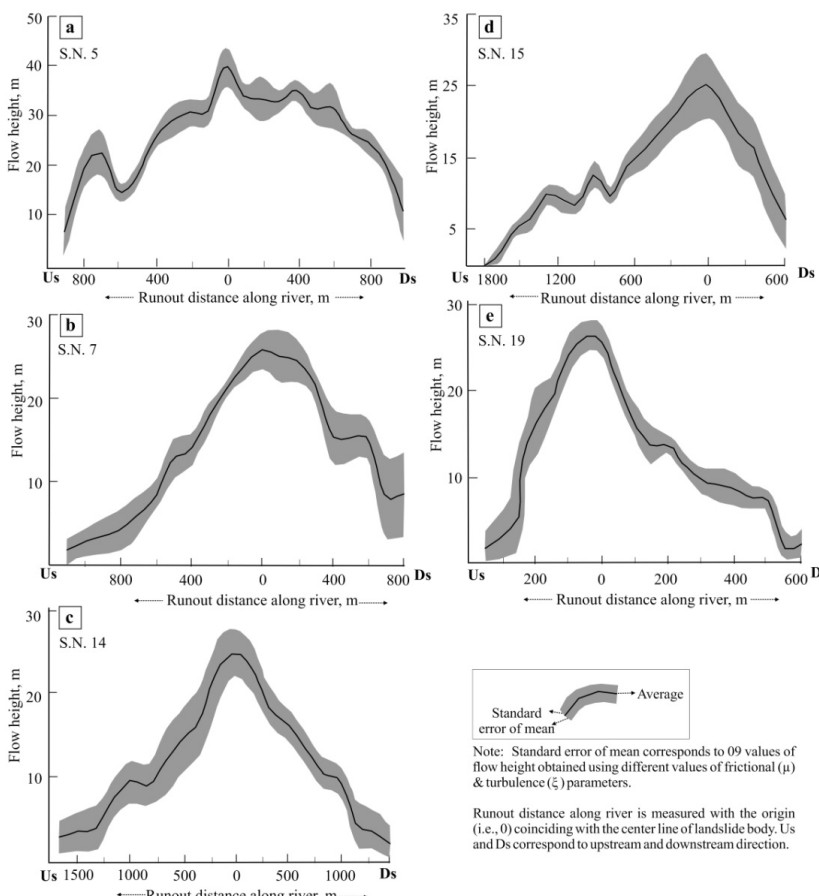

**Fig. 14** Results of run-out analysis at different values of μ and ξ. μ and ξ refer to coefficient of friction and turbulence, respectively.