# Peer review of "Inferring potential landslide damming using slope stability, geomorphic constraints and run-out analysis; case study from the NW Himalaya"

_Earth Surface Dynamics, 2020_

## Referee Comment (RC1) · Anonymous Referee #1 · 22 Oct 2020

I find this study is very relevant . The approach is quite good. Unstable landslides (out of 44) were identified through FEM and subsequently five landslides, those found unstable were further analysed for its blockage potential using a debris flow model. MOI and HDSI are used to evaluate potential of landslide damming. Many geotechnical parameters were estimated from field survey and laboratory analysis. This kind of investigation is quite less in literature although previously attempted by these authors for one landslide. I have some minor comments.

line 45 - Rapid Mass Movement Simulation (RAMMS) Line 102 - Do you mean KCF

is a splay fault of KF? Line 120 - It is a complex sentence. Pl. modify it. Line 161 - Pl. discuss briefly the spatial variability of compressional and extensional regime here. Line 203 - whether width of dammed valley is measured at full reservoir level? Runout analysis - This analysis was performed using RAMMS. The method and parameters are fairly well discussed. I missed your explanation w.r.t. release area. Pl. describe. I think you have assumed the flow as block release. Is there any chance of Channelised flow also? Line 256 - Since you have mentioned that majority of landslides are debris slides, pl. explain how the runout analysis, which is mainly done for debris flow, is valid in your study. Lie 415 - Your previous publication on Urni landslide gives a different flow height. Can you explain? Line 469 - What do you mean by strong / weak lithology. I suggest to use a technical terms here. 'therefore' is repeated. Table 1 - Have you assumed uniform thickness while estimating volume from area? How can you say that area measurement has error of 1.06% due to measurement from Google Earth image?

---

## Referee Comment (RC2) · Anonymous Referee #2 · 16 Dec 2020

Paper leaves the dual impression. Authors present a lot of data, numerous characteristics and parameters. At the same time it is difficult to understand if ladslides at the studied sites had occur already or they are just expected. How one can confirm that the parametrs of the assumed landslide dams are estimated correctly or not? Large parts of the text with numerous quantitative parameters can be replaced by tables that will be much easier to follow. I would suggest to rework the paper. First - clearly indicate what landslide that you mentioned had really occurred and what is just an unstable slope that might fail. It is especially important for rock avalanches - before such type of

landslide occur we cannot ne sure that it will not move just as a rockslide. Second - it will be very useful to analyze at least several case studies of past real river-damming landslides that can be used as a ground truth to check the reliability of the proposed approach.

---

## Author Comment (AC2) · 31 Dec 2020

v.chauhan777@gmail.com Received and published: 31 December 2020

**Response to Anonymous Referee #2**

Comment 1: Paper leaves the dual impression. Authors present a lot of data, numerous characteristics and parameters. At the same time it is difficult to understand if landslides at the studied sites had occurred already or they are just expected.

Response: We appreciate the critical, yet constructive, evaluation by the reviewer. The size of the data that we have included in our work reflects our comprehensive methodology. We would like to clarify that the landslides have already occurred but many of these are still active and result into frequent failures throughout the year. Stability evaluation is performed in the study to determine the existing stability regime of the landslide slopes through Factor of safety and displacement (Griffiths and lane, 1999). Later, these landslide slopes were categorized into unstable and meta-stable category based on their existing factor of safety.

Comment 2: How one can confirm that the parameters of the assumed landslide dams are estimated correctly or not?

Response: We are thankful for such perceptive query. We have used following parameters to evaluate the possibility of landslide damming; Landslide volume, Dam volume, Width of valley, Upstream catchment area and Local slope gradient of river channel. As stated in the MS, though the resultant (post failure) dam volume could be higher or lower than the landslide volume owing to the slope entrainment, rockmass fragmentation, retaining of material at the slope, and washout by the river (Hungr and Evans 2004; Dong et al. 2011). Exact influence of all these controlling factors on the volume difference can't be ascertained due to following reasons;

Slope entrainment, rock mass fragmentation and retaining of material at the slope will depend on the surface runoff, random joints/fractures on the slope surface, friction of slope the surface, turbulence of the moving mass on slope material and pore water pressure regime (Hungr and Evans 2004; Dong et al. 2011; Cui et al. 2019; Scott and Wohl, 2019). Since surface runoff, turbulence of moving mass and pore water regime are bound to change spatio-temporally, exact estimation seems difficult with available techniques/theories.

Washout of the failed material by the river will depend upon the river discharge. It is to mention that the Satluj River discharge is highly affected by the Western Disturbance and Indian Summer Monsoon induced precipitations, which have shown spatio-temporal variation (Gadgil et al. 2007; Wulf et al. 2012). Since the river is also subjected to many artificial dams (for hydroelectric power) in the upstream direction, river discharge might also get affected by these mega barriers (Kumar and Katoch, 2016). Thus, exact estimation of washout quantity is also difficult to ascertain.

Therefore, dam volume is assumed to be equal to landslide volume for worst case scenario. Similar assumption has also been made in other studies (Canuti et al., 1998) since the existing understanding of such landslide damming studies lack any exact relation between landslide volume and dam volume. Further, the main idea of the study is to predict potential landslide damming sites where damming is yet to takes place, except Urni landslide where it already occurred partially in year 2013 (Kumar et al. 2019a).

For the estimation of width of the valley, upstream catchment area and slope gradient, we used CartoSat-1 panchromatic imagery (spatial resolution 2.5m) and the DEM (spatial resolution 10m) constructed using stereo pairs. These Cartosat-1 imageries have also been evaluated for the morphometric measurement to determine their accuracy (Kandrika and Dwivedi, 2013).

Thus, though exact estimation of all these parameters can't be made due to the aforementioned limitations, we utilized available resources/theory. We are hopeful that the reviewer will perceive our justification rational enough for the query.

Comment 3: Large parts of the text with numerous quantitative parameters can be replaced by tables that will be much easier to follow. I would suggest reworking the paper.

Response: As per the suggestion, we shall be adding a table in the revised MS incorporating all details like landslides dimension, factor of safety, geomorphic indices output for each landslide. However, we are of understanding that the data mentioned in the text is required to justify the proposed approach and interlinked nature.

Comment 4: Clearly indicate what landslide that you mentioned had really occurred

СЗ

and what is just an unstable slope that might fail. It is especially important for rock avalanches - before such type of landslide occur we cannot be sure that it will not move just as a rockslide.

Response: As per the suggestion, we shall be clearly mentioning the state of slopes in the "Study Area" and "Results" sections. However, we would like to mention that we have stated in the "Study Area" section that it covers forty-four (44) 'active' landslides (20 debris slides, 13 rock falls, and 11 rock avalanches) along the study area that have been mapped recently by Kumar et al. (2019b). Further, we are of understanding that a clarification is required to be mentioned in the revised MS about the difference between landslide slope and unstable state. When we used the word "active landslide", it refers to the fact that hillslope is still subjected to slope failures caused by various factors. It, of course, doesn't mean that the entire hill slope has moved down. As we understand the word "landslide" can be perceived in following three ways; pre-failure deformations, failure itself, and post-failure displacement (Terzaghi 1950; Skempton and Hutchinson, 1969; Cruden & Varnes, 1996; Hungr et al., 2014). Landslide slopes in our study pertains to the post-failure state that are categorized into "unstable" and "meta-stable" stages based on their existing factor of safety. Furthermore, if an active landslide slope is not categorized as "unstable" in our study, it means that its existing slope geometry provides it a "meta-stable" stage that might transform into unstable stage with time due to stability controlling parameters (explained in the Sec. 4.1 in the MS).

Comment 5: It will be very useful to analyse at least several case studies of past real river-damming landslides that can be used as a ground truth to check the reliability of the proposed approach.

Response: We are thankful to the reviewer for the suggestion. We would like to clarify that our study attempts a predictive approach unlike the post-dam formation studies (detailed review in Fan et al. 2020).

In this predictive approach, at first the stability evaluation of active landslides is performed to determine their existing failure potential. Spatio-temporal regime of the failure triggering factors like earthquakes and rainfall is explored to infer the triggering possibility. Later, widely used geomorphic indices are used to find out those landslides that may result into damming of the river. These landslides are further evaluated using the run-out modelling to understand the response of failed slope material in the river channel and hence in the valley. Thus, the whole approach aims to find those slopes that will contribute to the damming, in case of slope failure.

We acknowledge the necessity of validation of the proposed approach, as suggested by the reviewer. Therefore, we would like to state that we have validated our predictive sites with the field observations (Fig. I, II) in the study area where damming has occurred in the past. Sedimentological analysis by other researchers has also confirmed the landslide damming events in the geological past at the region containing our predictive sites (Sharma et al. 2017). This approach has also been applied recently at an already dammed (partially) site where we predicted complete blockade of the river and consequent damage to the nearby bridge in case of further failure (Kumar et al. 2019a). As predicted, further failure blocked the river and damaged the bridges (https://timesofindia.indiatimes.com/city/shimla/nh-5-remains-blocked-dueto-landslides-in-himachal-pradesh/articleshow/74613645.cms, retrieved on 24th Dec. 2020).

Though we have provided the field examples (Fig. I, II), we can't deny the significance of multiple case studies involving existing damming sites, as suggested by the reviewer. However, it will require the back analysis of the following parameters to apply our proposed approach;

Back analysis of the damming volume to reconstruct the landslide volume. As mentioned in the response of the Comment 2, there are several uncertain spatio-temporal factors that play a crucial role between landslide volume and dam volume. The Exact response of these factors can't be ascertained at present.

Back analysis of the failed slope topography to reconstruct the pre-failure slope stability model. It will require the adjustment of landslide area (not the volume because we have performed 2D analysis). Such readjustment of the landslide area to pre-failure state will also include uncertainty because there might be many episodes of failures (which are not dated/recorded) that resulted in final topography. Regional faults/lineaments also affect the slope topography and thus during slope topography reconstruction, lack of inclusion of this factor will surely affect the reconstruction.

We are hopeful that the reviewer will understand the merits and limitations of the predictive nature of the approach that we tried to present judiciously.

**References**

Canuti P, Casagli N, Ermini L (1998) Inventory of landslide dams in the Northern Apennine as a model for induced iňĆood hazard forecasting. In: Andah, K. (Eds.), Managing hydro-geological disasters in a vulnerable environment for sustainable development, National Research Council of Italy, UNESCO (IHP), Porano, pp. 189–202.

Cruden DM, Varnes DJ (1996) Landslides: investigation and mitigation. Chapter 3-Landslide types and processes. Transportation research board special report, (247).

Cui Y, Jiang Y, & Guo C (2019) Investigation of the initiation of shallow failure in widely graded loose soil slopes considering interstitial flow and surface runoff. Landslides, 16(4): 815-828.

Dong JJ, Tung YH, Chen CC, Liao JJ and Pan YW (2011) Logistic regression model for predicting the failure probability of a landslide dam. Engineering Geology 117(1): 52-61.

Fan X, Dufresne A, Subramanian SS, Strom, A, Hermanns R, Stefanelli CT., ... & Geertsema M (2020) The formation and impact of landslide dams–State of the art. Earth-Science Reviews, 203, 103116.

Gadgil S, Rajeevan M and Francis PA (2007) Monsoon variability: Links to major oscillations over the equatorial Pacific and Indian oceans. Current Science. 93(2):182-194.

Griffiths DV, Lane PA (1999) Slope stability analysis by finite elements. Geotechnique, 49(3): 387-403.

Hungr O, Leroueil S, Picarelli L (2014) The Varnes classification of landslide types, an update. Landslides 11 (2): 167-194.

Hungr O and Evans SG (2004) Entrainment of debris in rock avalanches: an analysis of a long run-out mechanism. Geological Society of America Bulletin 116(9-10): 1240-1252.

Kandrika S & Dwivedi RS (2013) Reclamative grouping of ravines using Cartosat-1 PAN stereo data. Journal of the Indian Society of Remote Sensing, 41(3): 731-737.

Kumar V, Gupta V, Jamir I, Chattoraj S L (2019a) Evaluation of potential landslide damming: Case study of Urni landslide, Kinnaur, Satluj valley, India. Geosci. Front. 10(2): 753-767.

Kumar V, Gupta V, Sundriyal YP (2019b) Spatial interrelationship of landslides, lithotectonics, and climate regime, Satluj valley, Northwest Himalaya. Geol. J. 54: 537–551.

Kumar D & Katoch SS (2016) Environmental sustainability of run of the river hydropower projects: A study from western Himalayan region of India. Renewable Energy, 93: 599-607.

Scott DN & Wohl EE (2019) Bedrock fracture influences on geomorphic process and form across process domains and scales. Earth Surface Processes and Landforms, 44(1): 27-45.

Sharma S, Shukla AD, Bartarya SK, Marh BS & Juyal N (2017) The Holocene floods and their affinity to climatic variability in the western Himalaya, India. Geomorphology, 290: 317-334.

Skempton AW, Hutchinson JN (1969) Stability of natural slopes and embankment foundations, state-of-the-art report. Proc. 7th Int. Conf. Soil Mech. Found. Eng. 291-340.

Terzaghi K (1950) Mechanism of Landslides, in: Application of Geology to Engineering Practice.

Wulf H, Bookhagen B and Scherler D (2012) Climatic and geologic controls on suspended sediment flux in the Sutlej River Valley, western Himalaya". Hydrology and Earth System Sciences 16(7): 2193-2217.

Fig. Captions

Fig. I (As Fig. 10 in the Manuscript) Field signatures of the landslide damming near Akpa\_III landslide. (a) Upstream view of Akpa landslide with lacustrine deposit at the left bank; (b) enlarged view of lacustrine deposit with arrow indicating lacustrine sequence; (c) alternating fine-coarse sediments. F and C refer to fine (covered by yellow dashed lines) and coarse (covered by green dashed lines) sediments, respectively.

Fig. II (As Fig. 2 in Kumar et al. 2019a) Field photographs. (a) Front face of the Urni landslide; (b) and (c) are upstream and downstream view of the valley from the landslide location; (d) and (e) denote river damming during year 2013 and 2016, respectively; (f) slope in the opposite side. Red circles denote relative size by encircling heavy truck in 2b, 2e and tunnel outlet building (6 m x 4 m) in 2c.

Fig. 1. Fig I (As Fig. 10 in the Manuscript)

---

## Author Response (AR1)

**RESPONSE TO THE REFEREES**

**General**

We are thankful to Prof. X. Fan (Associate Editor) and two anonymous reviewers for their critical yet constructive comments that helped to improve the MS.

**Referee #1**

**Comment 1**: I find this study is very relevant. The approach is quite good. Unstable landslides (out of 44) were identified through FEM and subsequently five landslides, those found unstable were further analysed for its blockage potential using a debris flow model. MOI and HDSI are used to evaluate the potential of landslide damming. Many geotechnical parameters were estimated from field survey and laboratory analysis. This kind of investigation is quite less in literature although previously attempted by these authors for one landslide. I have some minor comments.

**Response**: We are grateful for the constructive comments/suggestions by the reviewer. We are also pleased that the reviewer perceives it as a valuable contribution. Below are our responses that are considered in the revised manuscript.

**Comment 2:** Line 45 - Rapid Mass Movement Simulation (RAMMS)

**Response**: We apologize for the typing error. As per the suggestion, it is corrected in the revised MS.

**Comment 3:** Line 102 - Do you mean KCF is a splay fault of KF?

**Response**: In our manuscript, we have stated that the Kaurik-Chago Fault (KCF) is subjected to the Karakoram Fault (KF). However, it does not intend to imply that the KCF is a splay fault of the KF. The KCF is an N-S oriented trans-tensional rift fault across the Himalayan strike that has been observed to extend to the north right up to the strike-slip Karakoram Fault. The Karakoram Fault follows the Himalayan strike in the NW Himalaya. The word "subjected" is used in the manuscript because the Kaurik- Chago Fault has been observed to differentiate between the NW and SE part of the Karakoram Fault that comprises different slip rates (Kundu et al. 2014). Nonetheless, to avoid the confusion, we have rephrased the sentence.

**Comment 4:** Line 120 - It is a complex sentence. Pl. modify it.

**Response**: As per the suggestion, it is modified in the revised MS for the further clarification.

**Comment 5**: Line 161 - Pl. discuss briefly the spatial variability of compressional and extensional regime here.

**Response**: The spatial variability of the compressional and extensional regime has been mentioned on the basis of observations of Vannay et al. (2004). As per the suggestion, it is being described as follows;

The study area in the Tethyan Sequence (TS) region has been observed to possess the NW-SE directed extensional regime based on the slickensides present on the brittle ductile structures. The Sangla detachment (SD) fault has been observed to comprise two regimes belonging to two different deformation phases. Earlier one corresponds to compression due to foreland thrusting whereas, later one corresponds to extension as evident from normal drag shear bands (Grasemann et al. 2003). The structural features in the Higher Himalaya Crystalline (HHC) reveal spatial variability of compression and extension regime. The structures in the upper part of the HHC are influenced by east directed extension along the SD fault. The lower part, however, comprises signs of SW directed compression along the Main Central Thrust (MCT). The structures in the Main Central Thrust (MCT) region have been observed to consist of a compressional regime, later superimposed by an extensional regime. In contrast to the HHC, structures in the Lesser Himalaya Crystalline (LHC) don't comprise any phase of the extensional regime and are influenced by the compressional regime. Based on the orientation of slickensides, kink bands, and other features, Vannay et al. (2004) observed SSW directed compressional regime in the Munsiari Thrust (MT) region. In the Lesser Himalaya Sequence (LHS) region, SW directed compressional regime has been observed on the basis of SW verging folds, crenulation cleavage, and other features. The same explanation is briefly added in the revised manuscript as per the suggestion.

**Comment 6:** Line 203 - whether width of dammed valley is measured at full reservoir level?

**Response**: We would like to clarify that the phrase "width of dammed valley" corresponds to the actual width of the section of the valley where damming is supposed to occur. For further clarification, the phrase "width of dammed valley" in the manuscript is revised to "width of the valley".

**Comment 7:** Runout analysis - This analysis was performed using RAMMS. The method and parameters are fairly well discussed. I missed your explanation wrt. release area. Pl. describe.

**Response:** We are thankful to the referee for pointing out this crucial aspect of the runout analysis. There are two possible ways to simulate the run-out event i.e., release area (for unchanneled flow or block release) and hydrograph (for channeled flow) concept. The channelized flow concept, however, requires spatial-temporal information of discharge at these flow channels (Rickenmann et al. 1999; RAMMS v.1.7.0). During the field visits, we did not find specific flow channels (or gullies) on the slope of landslides except few centimeters deep seasonal flow channels for S. N. 5 and S.N. 15 landslides (Table 1). However, the data pertaining to spatial-temporal information of discharge at these two landslides were not available. Therefore, we have chosen the release area concept because it is more appropriate when the flow

path (e.g. gully) is uncertain and its possible discharge on the slope is unknown. As per the suggestion, this explanation is added to the revised manuscript.

**Comment 8:** I think you have assumed the flow as block release. Is there any chance of Channelized flow also?

**Response**: We agree with the referee that we have considered the flow as a block release. As elaborated in the response to the previous comment, most of the landslides don't comprise specific flow channels except S. N. 5 and S.N. 15 landslides (Table 1). Though the possibility of channelized flow at these two landslides can't be denied, the data pertaining to spatial-temporal information of discharge at these two landslides were not available. We are hopeful that in further studies in the future, such data would be attempted to analyze.

**Comment 9:** Line 256 - Since you have mentioned that majority of landslides is debris slides, pl. explain how the runout analysis, which is mainly done for debris flow, is valid in your study.

**Response:** We are of understanding that the debris flow is a stage of debris-laden landslide that under excessive saturation results in the discharge of poorly sorted sediments (or debris) with varying velocity and pressure. Since the majority of the landslides are debris slides, as rightly pointed out by the referee, having unconsolidated poorly sorted overburden, there is a high probability that these debris-laden landslides will transform into debris flow during extreme rainfall events (Embley, 1976; Hungr et al. 2005; Jakob et al. 2005). Further, the study area has been witnessing enhanced rainfall since 2010 and subsequent flash floods (Fig. 11 in the Manuscript), run-out evaluation of the debris slides becomes more crucial. We are hopeful that the referee is convinced with our rational attempt of explanation.

**Comment 10**: Line 415 - Your previous publication on Urni landslide gives a different flow height. Can you explain?

**Response**: We acknowledge that the previous publication involving the Urni landslide had a different flow height than the one mentioned in the present study. The reason for this difference pertains to the following input parameters; friction, turbulence, and depth. The previous study utilized single values of friction and turbulence, whereas in the present study we have used a range (9 sets of values) of these parameters to minimize the possible uncertainty in output (sec. 3.5 in the manuscript). Further, we have been more conservative in the selection of depth in the present study because these landslides are relatively deep in nature and we are of the understanding that during slope failure, irrespective of the type of trigger, entire loose material might not slide down. Therefore, the depth of the landslide is taken as only 1/4 (thickness) in the run-out calculation.

**Comment 11:** Line 469 - What do you mean by strong / weak lithology. I suggest using a technical term here.

**Response:** As per the suggestion, it is replaced in the revised MS with the proper lithology term.

**Comment 12:** 'therefore' is repeated

**Response**: We apologize for the typing error. It is removed in the revised manuscript.

**Comment 13**: Table 1 - Have you assumed uniform thickness while estimating volume from area?

**Response:** We would like to clarify that we have not assumed uniform thickness for the volume calculation. These landslides were recently mapped by our team and a detailed procedure has been mentioned in Kumar et al. (2019b). We are quoting the same here for clarification.

"The landslide dimension mapping was performed using high resolution GE Imagery, and their locations were verified during field investigation. The uncertainty in the landslide dimension caused by measurement in GE was determined by comparing the known distances in the study area with the measured ones in GE. The known distances were obtained from the Survey of India toposheets (53/I/10, 53/I/6, 53/I/2, 53/E/4, and 53/E/11). A difference of 1.06% was noted between known distances (from toposheets) and measured ones in the GE. Landslide dimension was characterized using the area (total disturbed area), shape (length and width), and volume. Approximate thickness and area of landslides were used to determine the volume. The thickness of individual landslides was ascertained in the field investigation, as also practiced by Larsen and Torres-Sanchez (1998) and Guzzetti et al. (2009)."

**Comment 14**: How can you say that area measurement has error of 1.06% due to measurement from Google Earth image?

**Response**: As mentioned in the response to the previous comment, an error of 1.06% was noted between the known distances (from toposheets) and measured ones in Google Earth (GE) imagery. The known distances were obtained from the Survey of India toposheets (53/I/10, 53/I/6, 53/I/2, 53/E/4, and 53/E/11).

We tried our best to rationally convince the referee with our explanations and we are hopeful that these responses will be received constructively.

**Referee #2**

Comments:

**Comment 1**: Paper leaves the dual impression. Authors present a lot of data, numerous characteristics and parameters. At the same time it is difficult to understand if landslides at the studied sites had occurred already or they are just expected.

**Response**: We appreciate the critical, yet constructive, evaluation by the reviewer. The size of the data that we included in our work reflects our comprehensive methodology. We would like to clarify that the landslides have already occurred but many of these are still active and result into frequent failures throughout the year. Stability evaluation is performed to determine the existing stability regime of the landslide slopes through the Factor of safety (Griffiths and lane, 1999) and displacement. Later, these landslide slopes were categorized into unstable and meta-stable class based on their existing factor of safety.

**Comment 2**: How one can confirm that the parameters of the assumed landslide dams are estimated correctly or not?

**Response:** We are thankful for such perceptive query. We have used following four parameters to evaluate the possibility of potential landslide dams; *Landslide volume, Dam volume, Width of valley, Upstream catchment area, Local slope gradient of river channel.*
As stated in the MS, the resultant (post-failure) dam volume could be higher or lower than the landslide volume owing to the slope entrainment, rock mass fragmentation, retaining of material at the slope, and washout by the river (Hungr and Evans 2004; Dong et al. 2011). Exact influence of all these controlling factors on the volume difference can't be ascertained due to the following reasons;

- Slope entrainment, rock mass fragmentation, and retaining of material at the slope will depend on the surface runoff, random joints/fractures on the slope surface, friction of slope the surface, turbulence of the moving mass on slope material, and pore water pressure regime (Hungr and Evans 2004; Dong et al. 2011; Cui et al. 2019; Scott and Wohl, 2019). Since the surface runoff, turbulence of moving mass, and pore water regime are bound to change spatio-temporally, exact estimation seems difficult with the available techniques/theories.

- Washout of the failed material by the river will depend upon the river discharge. It is to mention that the Satluj River discharge is highly affected by the Western Disturbance and Indian Summer Monsoon induced precipitations, which have shown spatio-temporal variation (Gadgil et al. 2007; Wulf et al. 2012). Since the river in the study area is also subjected to many artificial dams (for hydroelectric power) in the upstream, river discharge might also get affected by these mega barriers (Kumar and Katoch, 2016). Thus, the exact estimation of washout quantity is difficult to ascertain.

- Therefore, dam volume is assumed to be equal to the landslide volume for the worst case scenario. Similar assumption has also been made in other studies (Canuti et al., 1998) since the existing understanding of such landslide damming studies lack any exact relation between landslide volume and dam volume. Further, the main idea is to predict the potential landslide damming sites where damming is yet to takes place, except the Urni landslide where it already occurred partially in year 2013 (Kumar et al. 2019a).

- For the estimation of width of the valley, upstream catchment area, and slope gradient, we used CartoSat-1 panchromatic imagery (spatial resolution 2.5m) and the DEM (spatial resolution 10m) constructed using the stereo pairs of CartoSat-1. These Cartosat-1 imageries have also been evaluated for the morphometric measurement to determine their accuracy (Kandrika and Dwivedi, 2013).

Thus, though the exact estimation of all these parameters can't be made due to various limitations (mentioned above), we utilized available resources/theory. We are hopeful that the reviewer will find our justification rational enough to support the findings.

**Comment 3**: Large parts of the text with numerous quantitative parameters can be replaced by tables that will be much easier to follow. I would suggest reworking the paper.

**Response:** We understand the reviewer's perception about the size of the data mentioned in the MS. Therefore, as per the insightful suggestion, we have added a supplementary table 3 in our openly accessed data repository (Kumar et al. 2020) related to this study. This table comprises all the details like landslides no., dimension, position, litho-tectonic affiliation, factor of safety, geomorphic indices output for each landslide. However, we are of understanding that the data mentioned in the text is required to justify the proposed approach and the interlinked nature of the study.

**Comment 4:** Clearly indicate what landslide that you mentioned had really occurred and what is just an unstable slope that might fail. It is especially important for rock avalanches - before such type of landslide occur we cannot be sure that it will not move just as a rockslide.

**Response:** As per the suggestion of the reviewer, we have clearly mentioned about the state of slopes in the "Discussion" section. We would also like to mention that we had stated in the "Study Area" section in the original MS that it covers forty-four (44) '*active*' landslides (20 debris slides, 13 rock falls, and 11 rock avalanches) along the study area that have been mapped recently by Kumar et al. (2019b).
Further, in view of the understanding that a clarification is required to be mentioned about the difference between landslide slope and unstable state, we have mentioned it in the 'Discussion'. When we used the word "active landslide", it refers to fact that hillslope is still subjected to slope failures caused by various factors. It, of course, doesn't mean that the entire hill slope has moved

down. As we understand the word "landslide" can be perceived in following three ways; pre-failure deformations, failure itself, and post-failure displacement (Terzaghi 1950; Skempton and Hutchinson, 1969; Cruden & Varnes, 1996; Hungr et al., 2014). Landslide slopes in our study pertains to the post-failure state that are categorized into "unstable" and "metastable" stages based on their existing factor of safety.

Furthermore, if an *active* landslide slope is not categorized as "unstable" in our study, it means that its existing slope geometry provides it a "meta-stable" stage that might transform into unstable stage with time due to stability controlling parameters (inferred in parametric analysis).

**Comment 5**: It will be very useful to analyze at least several case studies of past real river-damming landslides that can be used as a ground truth to check the reliability of the proposed approach.

**Response:** We are thankful to the reviewer for the suggestion. We would like to clarify that our study attempts a predictive approach unlike the post-dam formation studies (detailed review in Fan et al. 2020).

In this predictive approach, at first the stability evaluation of active landslides is performed to determine their existing failure potential. Spatio-temporal regime of the failure triggering factors like earthquakes and rainfall is explored to infer the triggering possibility. Later, widely used geomorphic indices are used to find out those landslides that may result into damming of the river. These landslides are further evaluated using the run-out modeling to understand the response of failed slope material in the river channel and hence in the valley. Thus, the whole approach aims to find those slopes that will contribute to the damming, in case of slope failure.

We acknowledge the necessity of validation of the proposed approach, as brought up by the reviewer. Therefore, we would like to state that we have validated our predictive sites with the field observations (Fig. 1, 2) in the study area where damming has occurred in the past. Sedimentological analysis by other researchers has also confirmed the landslide damming events in the geological past at the region containing our predictive sites (Sharma et al. 2017). This approach has also been applied recently at already dammed (partially) site where we predicted complete blockade of the river and consequent damage to the nearby bridge in case of further failure (Kumar et al. 2019a). As predicted, further failure blocked the river and damaged the bridges (https://timesofindia.indiatimes.com/city/shimla/nh-5-remains-blocked-due-to-landslides-in-himachal-pradesh/articleshow/74613645.cms, retrieved on 24th Dec. 2020).

Though we have provided the field examples (Fig. I, II), we can't deny the significance of multiple case studies involving the existing damming sites, as suggested by the reviewer. However, it will require the back analysis of the following parameters to apply our proposed approach;

- Back analysis of the damming volume to reconstruct the landslide volume. As mentioned in the response of the Comment 2, there are several uncertain spatio-temporal factors that play crucial role between landslide volume and dam volume. Exact response of these factors can't be ascertained at present.
- Back analysis of the failed slope topography to reconstruct the pre-failure slope stability model. It will require the adjustment of landslide area (not the volume because we have performed 2D analysis). Such readjustment of the landslide area to pre-failure state will also include uncertainty because there might be many episodes of failures (which are not dated/recorded) that resulted in final topography. Regional faults/lineaments also affect the slope topography and thus during slope topography reconstruction, lack of inclusion of this factor will surely affect the reconstruction.

We are hopeful that the reviewer will understand the merits and limitations of the predictive nature of our approach that we tried to present judiciously.

**Author's changes in the manuscript**

(It is to note that the following line numbers refer to the document in track change: All mark up mode)

Line no. 45 (Page No. 2): In response to the comment 2 (Referee#1), the typing error has been corrected.

Line no. 120-125 (Page No. 5): In response to the comment 4 (Referee#1), the paragraph has been revised.

Line no. 159-167 (Page No. 6): In response to the comment 5 (Referee#1), details regarding the stress regime have been added.

Line no. 207 (Page No. 8): In response to the comment 6 (Referee#1), the phrase 'width of dammed valley' has been revised to 'width of the valley'.

Line no. 254-261 (Page No. 9): In response to the comment 7 (Referee#1), details regarding the release area have been added.

Line no. 470- 471,476-478 (Page No. 16): In response to the comment 11 (Referee#1), the proper lithological terms are used.

Line no. 478 (Page No. 16): In response to the comment 12 (Referee#1), a repetition has been removed.

Line no. 543-553 (Page No. 18-19): In response to the comment 4 (Referee#2), a clarification is added in the 'Discussion' section.

Line no. 551-553 (Page No. 19): In response to the comment 3 (Referee#2), we have added a supplementary table 3 in our openly accessed data repository (Kumar et al. 2020) related to this study.

**References**

Canuti, P., Casagli, N. and Ermini, L. (1998). Inventory of landslide dams in the Northern Apennine as a model for induced flood hazard forecasting. In: Andah, K. (Eds.), Managing hydro-geological disasters in a vulnerable environment for sustainable development, National Research Council of Italy, UNESCO (IHP), Porano, pp. 189–202.

Cruden, D.M., Varnes, D.J. (1996). Landslides: investigation and mitigation. Chapter 3-Landslide types and processes. Transportation research board special report, (247).

Cui, Y., Jiang, Y., & Guo, C. (2019). Investigation of the initiation of shallow failure in widely graded loose soil slopes considering interstitial flow and surface runoff. Landslides 16(4), 815-828.

Dong, J.J., Tung, Y.H., Chen, C.C., Liao, J.J. and Pan, Y.W. (2011). Logistic regression model for predicting the failure probability of a landslide dam. Engineering Geology 117(1), 52-61.

Embley, R. W. (1976). New evidence for occurrence of debris flow deposits in the deep sea. Geology 4(6), 371-374.

Fan, X., Dufresne, A., Subramanian, S. S., Strom, A., Hermanns, R., Stefanelli, C. T., ... & Geertsema, M. (2020). The formation and impact of landslide dams–State of the art. Earth-Science Reviews 203, 103116.

Gadgil, S., Rajeevan, M. and Francis, P.A. (2007).Monsoon variability: Links to major oscillations over the equatorial Pacific and Indian oceans. Current Science 93(2):182-194.

Griffiths, D.V., Lane, P.A. (1999). Slope stability analysis by finite elements. Geotechnique 49(3), 387-403.

Guzzetti, F., Ardizzone, F., Cardinali, M., Rossi, M., & Valigi, D. (2009). Landslide volumes and landslide mobilization rates in Umbria, central Italy. Earth and Planetary Science Letters, 279(3), 222–229.

Hungr, O., Leroueil, S., Picarelli, L. (2014). The Varnes classification of landslide types, an update. Landslides 11 (2), 167-194.

Hungr, O. and Evans, S.G. (2004). Entrainment of debris in rock avalanches: an analysis of a long run-out mechanism. Geological Society of America Bulletin 116(9-10), 1240-1252.

Hungr, O., McDougall, S., & Bovis, M. (2005). Entrainment of material by debris flows. In Debris-flow hazards and related phenomena. Springer, Berlin, Heidelberg.

Jakob, M., Hungr, O., & Jakob, D. M. (2005). Debris-flow hazards and related phenomena. Springer, Berlin, Heidelberg.

Kandrika, S., & Dwivedi, R. S. (2013). Reclamative grouping of ravines using Cartosat-1 PAN stereo data. Journal of the Indian Society of Remote Sensing, 41(3), 731-737.

Kumar, V., Gupta, V., Jamir, I., Chattoraj, S. L. (2019a). Evaluation of potential landslide damming: Case study of Urni landslide, Kinnaur, Satluj valley, India. Geosci. Front. 10(2), 753-767.

Kumar, V., Gupta, V., Sundriyal, Y.P. (2019b). Spatial interrelationship of landslides, litho-tectonics, and climate regime, Satluj valley, Northwest Himalaya. Geol. J. 54, 537–551.

Kumar, V., Jamir, I., Gupta, V. and Bhasin, R.K. (2020). Dataset used to infer regional slope stability, NW Himalaya. Mendeley Data. DOI: 10.17632/jh8b2rh8nz.1

Kumar, D., & Katoch, S. S. (2016). Environmental sustainability of run of the river hydropower projects: A study from western Himalayan region of India. Renewable Energy, 93, 599-607.

Kundu, B., Yadav, R.K., Bali, B.S., Chowdhury, S. and Gahalaut, V.K. (2014).Oblique convergence and slip partitioning in the NW Himalaya: Implications from GPS measurements. Tectonics 33: 2013-2024

Larsen, M. C., & Torresâ˜ARˇ Sánchez, A. J. (1998). The frequency and distribution of recent landslides in three montane tropical regions of Puerto Rico. Geomorphology 24(4), 309–331

Rickenmann, D. (1999). Empirical relationships for debris flows. Natural hazards 19(1), 47-77.

Scott, D. N., & Wohl, E. E. (2019). Bedrock fracture influences on geomorphic process and form across process domains and scales. Earth Surface Processes and Landforms 44(1), 27-45.

Sharma, S., Shukla, A. D., Bartarya, S. K., Marh, B. S., & Juyal, N. (2017). The Holocene floods and their affinity to climatic variability in the western Himalaya, India. Geomorphology 290, 317-334.

Skempton, A.W., Hutchinson, J.N. (1969). Stability of natural slopes and embankment foundations, state-of-the-art report. Proc. 7th Int. Conf. Soil Mech. Found. Eng. 291-340.

Terzaghi, K. (1950). Mechanism of Landslides, in: Application of Geology to Engineering Practice.

Vannay, J.C., Grasemann, B., Rahn, M., Frank, W., Carter, A., Baudraz, V. and Cosca, M. (2004)."Miocene to Holocene exhumation of metamorphic crustal wedges in the NW

Wulf, H., Bookhagen, B. and Scherler, D. (2012).Climatic and geologic controls on suspended sediment flux in the Sutlej River Valley, western Himalaya". Hydrology and Earth System Sciences 16(7): 2193-2217.

**Fig. Captions**

Fig. I (As Fig. 10 in the Manuscript) Field signatures of the landslide damming near Akpa_III landslide. (a) Upstream view of Akpa landslide with lacustrine deposit at the left bank; (b) enlarged view of lacustrine deposit with arrow indicating lacustrine sequence; (c) alternating fine-coarse sediments. F and C refer to fine (covered by yellow dashed lines) and coarse (covered by green dashed lines) sediments, respectively.

Fig. II (As Fig. 2 in Kumar et al. 2019a) Field photographs. (a) Front face of the Urni landslide; (b) and (c) are upstream and downstream view of the valley from the landslide location; (d) and (e) denote river damming during year 2013 and 2016, respectively; (f) slope in the opposite side. Red circles denote relative size by encircling heavy truck in 2b, 2e and tunnel outlet building (6 m x 4 m) in 2c.

---

## Editor Decision (ED1)

[revised manuscript text omitted]

|----------------|---------|---------------------------------------------------------|-------------------------------|--------------------|
|                | 524/253 |                                                         | 5 th Dec.
2010  | ~2.5 m             |
|                | 525/253 |                                                         | 16 th Dec.
2010 | ~2.5 m             |
| CADTOGAT       | 526/252 |                                                         | 18 th Oct.
2011 | ~2.5 m             |
| 1 stereo       | 526/253 | National Remote Sensing Center (NRSC), Hyderabad, India | 18 th Oct.
2011 | ~2.5 m             |
| innagery       | 527/252 |                                                         | 24 th Nov
.2010 | ~2.5 m             |
|                | 527/253 |                                                         | 27 th Dec.
2010 | ~2.5 m             |
|                | 528/252 |                                                         | 26 th Nov.
2011 | ~2.5 m             |

 Table 2 Details of the satellite imagery.

|                      | Material Criteria                                                                                                                                                                                        | Parameters                                         | Source                                                                                            |  |
|----------------------|----------------------------------------------------------------------------------------------------------------------------------------------------------------------------------------------------------|----------------------------------------------------|---------------------------------------------------------------------------------------------------|--|
|                      | Generalized Hoek & Brown (GHB) Criteria
(Hoek et al. 1995)                                                                                                                                            | Unit Weight, γ
(MN/m 3 )             | Laboratory analysis (UCS)                                                                         |  |
|                      | $\sigma_1 = \sigma_3 + \sigma_{ci} [m_b(\sigma_3/\sigma_{ci}) + s]^{\wedge} a$                                                                                                                           | Uniaxial Compressive Strength, $\sigma_{ci}$ (MPa) | (IS: 9143-1979)                                                                                   |  |
|                      | Here, $\sigma_1$ and $\sigma_3$ are major and minor effective principal stresses at failure; $\sigma_{ci}$ , compressive strength of intact rock: $m_{b_i}$ a reduced value of the material constant (m) | Rockmass modulus
(MPa)                          | Laboratory analysis
(Ultrasonic velocity test); Hoek                                           |  |
|                      | and is given by;                                                                                                                                                                                         | Poisson's Ratio                                    | and Diederichs (2006).                                                                            |  |
| ckma <del>ss –</del> | $m_b = m_i e^{[(GSI-100)/(28-14D]]}$                                                                                                                                                                     | Geological Strength
Index                       | Field observation and based on
recent amendments (Cai et al.                                   |  |
| Roc                  | s and a; constants for the rock mass given by the following relationships;                                                                                                                               | Material Constant                                  | 2007 and reference therein)                                                                       |  |
|                      | $s = e^{[(GSI - 100)/(9 - 3D)]}$                                                                                                                                                                         | (m i )                                  | Standard values
(Hoek and Brown 1997)                                                          |  |
|                      | $a = \frac{1}{2} + \frac{1}{6} \left[ \mathbf{e}^{\left[ -(\frac{\alpha 3}{15}) \right]} - \mathbf{e}^{\left[ -(\frac{\alpha 3}{3}) \right]} \right]$                                                    | m b                                     | GSI was field dependent. m: as                                                                    |  |
|                      | Here, D; a factor which depends upon the degree of disturbance to which the rock mass has been subjected                                                                                                 | S                                                  | per(Hoek and Brown 1997) and
D is used between 0-1 in view                                     |  |
|                      | by blast damage and stress relaxation. GSI (Geological Strength Index); a rockmass characterization parameter.                                                                                           | а                                                  | of rockmass exposure and                                                                          |  |
|                      |                                                                                                                                                                                                          | D                                                  | olasting.                                                                                         |  |
|                      | Barton-Bandis Criteria
(Barton and Choubey 1977; Barton and Bandis 1990)                                                                                                                       | Normal Stiffness, k n                   | E i is lab dependent.L and GSI
were field dependent. D is                           |  |
|                      | $\tau = \sigma_{n} \tan \left[ \phi_{r} + JRC \log_{10} \left( JCS / \sigma_{n} \right) \right]$                                                                                                         | (MPa/m)                                            | used between 0-1 in view of rockmass exposure and blasting.                                       |  |
|                      | Here, $\tau$ is joint shear strength; $\sigma_n$ , normal stress across joint; $\emptyset_r$ , reduced friction angle; JRC, joint roughness coefficient; JCS, joint compressive strength.                | Shear Stiffness , k s
(MPa/m)        | It is assumed as k n /10.
However, effect of denominator
is aslo obtainedthrough |  |
| Joint                | JRC is based on the chart of Barton and Choubey (1977); Jang et al. (2014).JCS was determined using                                                                                                      |                                                    | parameteric study.                                                                                |  |
|                      | following equation;
$log_{10}(JCS) = 0.00088 (R_1)(\gamma)+1.01$                                                                                                                                      | Reduced friction angle, $Ø_r$                      | Standard values (Barton and Choubey 1977).                                                        |  |
|                      | Here, $R_L$ isSchimdt Hammer Rebound value and $\gamma$ is unit weight of rock.                                                                                                                          | Joint roughness                                    | Field based data from
profilometer and standard
values from Barton and                      |  |
|                      | The JRC and JCS were used as $JRC_n$ and $JCS_n$ following the scale corrections observed by Barton and Choubey                                                                                          | coefficient, JRC                                   | Choubey (1977); Jang et al.
(2014).                                                            |  |

| Table 3 | Criteria | used in | the | Finite | Element | Method | (FEM) | analysis. |
|---------|----------|---------|-----|--------|---------|--------|-------|-----------|
|         |          |         |     |        |         |        |       |           |

|      | (1977) and reference therein and proposed by Barton
and Bandis (1982).
$JRC_n = [JRC(L/L_o)^{-0.02(JRC)}]$ $JCS_n = [JCS(L/L_o)^{-0.03(JRC)}]$ Here, Land L o are mean joint spacing in field and,
respectively. L has been suggested to be 10 cm                                                                                                                               | Joint compressive
strength, JCS (MPa)   | Empirical equation of Deere and
Miller (1966) relating Schimdt
Hammer Rebound (SHR)
values, $\sigma_{ci}$ and unit weight of
rock. SHR was field dependent. |
|------|-----------------------------------------------------------------------------------------------------------------------------------------------------------------------------------------------------------------------------------------------------------------------------------------------------------------------------------------------------------------------------------------------------|--------------------------------------------|-------------------------------------------------------------------------------------------------------------------------------------------------------------------------|
|      | Joint stiffness criteria
(Barton 1972)                                                                                                                                                                                                                                                                                                                                                           | Scale corrected, JRC n          |                                                                                                                                                                         |
|      | $\label{eq:kn} \begin{split} &k_n = (E_i^*E_m)/L^*(E_i - E_m) \\ &\text{Here, } k_n; \text{ Normal stiffness, } E_i; \text{ Intact rock modulus, } \\ &E_m; \text{ Rockmass modulus } L; \text{ Mean joint spacing.} \\ &E_m = (Ei)^*[0.02 + \{1 - D/2\}/\{1 + e^{(60 + 15^*D - GSI)/11}\}] \\ &\text{Here, } E_m \text{ is based on Hoek and Diederichs (2006) and reference therein} \end{split}$ | Scale corrected, JCS n
(MPa) | Empirical equation of Barton
and Bandis (1982).                                                                                                                      |
|      | Mohr-Coulomb Criteria                                                                                                                                                                                                                                                                                                                                                                               | Unit Weight (MN/m 3 )           | Laboratory analysis (UCS)
(IS: 2720-Part 4–1985; IS:
2720-Part 10-1991)                                                                                           |
| Soil | (Coulomb 1776; Mohr 1914)
$\boldsymbol{\tau} = \boldsymbol{C} + \boldsymbol{\sigma}  \boldsymbol{tan} \boldsymbol{\emptyset}$                                                                                                                                                                                                                                                                    | Young's Modulus, E i
(MPa)   | Laboratory analysis (UCS); IS:
2720-Part 10-1991.                                                                                                                    |
|      | Here, $\tau$ ; Shear stress at failure, C; Cohesion, $\sigma_n$ ; normal strength, Ø; angle of friction.                                                                                                                                                                                                                                                                                            | Poisson's Ratio                            | Standard values from Bowles (1996)                                                                                                                                      |
|      |                                                                                                                                                                                                                                                                                                                                                                                                     | Cohesion, C (MPa)                          | Laboratory analysis (Direct shear)                                                                                                                                      |
|      |                                                                                                                                                                                                                                                                                                                                                                                                     | Friction angle, Ø                          | (IS: 2720-Part 13- 1986)                                                                                                                                                |

| Landslide Material
type |            | Material
depth 1 , m | Friction coefficient 2 | Turbulence
coefficient 3 , m/sec 2 |
|----------------------------|------------|------------------------------------|-----------------------------------|-------------------------------------------------------------|
| Akpa                       | Gravelly   | 5                                  | μ=0.05, 0.1, 0.3                  | $\xi = 100, 200, 300$                                       |
| (S.N. 5)                   | sand       |                                    |                                   |                                                             |
| Baren Dogri                | Gravelly   | 1.25                               | $\mu = 0.05, 0.1, 0.4$            | $\xi = 100, 200, 300$                                       |
| (S.N. 7)                   | N. 7) sand |                                    |                                   |                                                             |
| Pawari                     | Gravelly   | 1.25                               | $\mu = 0.05, 0.1, 0.4$            | $\xi = 100, 200, 300$                                       |
| (S.N. 14)                  | sand       |                                    |                                   |                                                             |
| Telangi                    | Gravelly   | 6.25                               | $\mu = 0.05, 0.1, 0.4$            | $\xi = 100, 200, 300$                                       |
| (S.N. 15)                  | sand       |                                    |                                   |                                                             |
| Urni                       | Gravelly   | 2.5                                | μ=0.06, 0.1, 0.4                  | $\xi = 100, 200, 300$                                       |
| (S.N. 19)                  | sand       |                                    |                                   |                                                             |

1 Considering that fact that during slope failure, irrespective of type of trigger, entire loose material might not slide down, the depth is taken as only 1/4 (thickness) in the calculation.2 Since the angle of run-out track (slope and river channel) varied a little beyond the suggested range 2.8° -21.8° or  $\mu = 0.05$ -0.4 (Hungr et al., 1984; RAMMS v.1.7.0), we kept out input in this suggested range wherever possible to avoid simulation uncertainty. 3This range is used in view of the type of loose material i.e., granular in this study (RAMMS v.1.7.0).

**Table 4** Details of input parameters for run-out analysis. S.N. refers to serial number of landslides in Fig. 1.

---

## Author Response (AR2)

**Editor (Prof. Joshua West)'s comments:**

We are thankful to Prof. J. West for his insightful comments and suggestions that improved the manuscript in its final version. We have updated the MS as per the suggestions of Editor. Further, some queries of Editor are addressed below;

**Comment 1**: (Sec 4.4) I agree with the reviewer that it would help in a place like this to clarify a bit more in terms of observations of existing landslides vs. model predictions.

**Response:**
We acknowledge the necessity of validation of the proposed approach, as also brought up by the reviewer 2 in the comment 5 (In our response to reviewers).

As stated in the response, we have validated some of our predictive sites with the field observations in the study area where damming has already occurred in the past. Sedimentological analysis by other researchers has confirmed the landslide damming events in the geological past at the region containing our predictive sites (Sharma et al. 2017). This approach has also been applied recently at already dammed (partially) site where we predicted complete blockade of the river and consequent damage to the nearby bridge in case of further failure (Kumar et al. 2019a). As predicted, further failure blocked the river and damaged the bridges (https://timesofindia.indiatimes.com/city/shimla/nh-5-remains-blocked-due-to-landslides-in-himachal-pradesh/articleshow/74613645.cms, retrieved on 24th Dec. 2020).

Though we have provided the field examples, we cannot deny the significance of multiple case studies involving the existing damming sites, as suggested by the reviewer. However, it will require the back analysis of the following parameters to apply our proposed approach;

- Back analysis of the damming volume to reconstruct the landslide volume. As mentioned in the response of the Comment 2 (Reviewer 2), there are several uncertain spatio-temporal factors that play crucial role between landslide volume and dam volume. Exact response of these factors can't be ascertained at present.
- Back analysis of the failed slope topography to reconstruct the pre-failure slope stability model. It will require the adjustment of landslide area (not the volume because we have performed 2D analysis). Such readjustment of the landslide area to pre-failure state will also include uncertainty because there might be many episodes of failures (which are not dated/recorded) that resulted in final topography. Regional faults/lineaments also affect the slope topography and thus during slope topography reconstruction, lack of inclusion of this factor will surely affect the reconstruction.

    Nonetheless, in view of the Editor's suggestions, we plan to overcome this research gap in the future prospects.

**Comment 2** (Sec 5.0): I am surprised by this statement, since I expect this region (with rapid exhumation) to have steeper slopes.

**Response:** We agree with the Editor's remark regarding the relationship of rapid exhumation and steeper slopes in the Higher Himalaya Crystalline (HHC) region. However, the statement in the MS intends to avoid such generalization in the HHC because the landslides (S.N. 7,14,15) are different in type and situated in different parts of the HHC.

The HHC region in the study area is found to possess two sub-regions that can be classified on the basis of lithology, climate, Normalized Difference Vegetation Index (NDVI), landslide type, and geomorphology (Kumar et al. 2019b).

The northern part comprises migmatitic gneiss and lies in the proximity of the SD (normal fault), whereas the southern part belongs to kyanite-sillimanite gneiss in the equal vicinity of the SD and MCT (thrust fault). These sub-regions also experience a spatial transition of climate from semi-humid (southern part) to semi-arid (northern part). This climatic transition is supported by the NDVI variation as it changes from 0.6 to 0.4 from the MCT to SD, respectively. The northern region comprises mainly bedrock landslides (rockfall, rock avalanche), whereas the southern region is dominated by the debris slides. The S.N. 7 is a rock avalanche type landslide, situated in the northern part of the HHC, whereas S.N. 14, 15 are debris slides that are situated in the southern part of the HHC. The northern region coexists with narrow, deep gorges and high topographic relief, whereas the southern region belongs to relatively wide valley and orographic frontal position that imply towards relatively more weathering in southern part of the HHC.

**Comment 3** (Sec 5.0): What about slopes that may be unstable under heavy rainfall or seismic shaking, but have not yet failed, or where failures are not visible because vegetation has re-grown?

**Response:** This question comprises two scenarios; (1) slope that may be unstable under heavy rainfall or seismic shaking, but have not yet failed, and (2) slopes where failures are not visible because vegetation has re-grown.

The first scenario is perhaps difficult to comprehend at present because it is subjected to time and uncertain triggering events (rainfall or seismic shaking). Since the hillslopes in such region are always subjected to the weathering process that varies spatio-temporally, it might be difficult to identify such slopes until any visible failure signs (cracks and/or movement of loose material) occurs. However, we would like to explore this aspect using the combination of InSAR and numerical simulation in the future prospects.

The second scenario belongs to the possibility of vegetation growth on the failed slopes. Though the field visits were performed in different seasons to eliminate such possibility, this scenario might

exist, particularly in the Lesser Himalaya Crystalline (LHC) and Lesser Himalaya Sequence (LHS). However, the landslides in the LHC and LHS are mostly rockfall/rock avalanche type because of the deep gorge setting, whereas the vegetation growth generally requires the debris laden hillslopes. Nevertheless, a statement in the 'Discussion' (Sec 5.0) has been mentioned about such a possibility to be covered in the future research prospects. The Higher Himalaya Crystalline (HHC) and the Tethyan Sequence (TS) region are subjected to the semi-humid to semi-arid climate and hence the vegetation type is mostly scattered trees/shrubs. Therefore, the second scenario might not be applicable here.

**Comment 4** (Fig. 8)**:** Perhaps explain here that events in the instability domain are not expected to form landslide dams?

**Response:** We would like to mention that the instability domain pertains to the 'durability' of the dams that have (or will) form in case of slope failure. As shown in Fig. 8a, there are five landslides that may form the landslide dams (in Formation domain) and twenty-four landslides, which will not form (Non-formation domain). Later, instability domain (Fig. 8b) highlights the 'durability' of those dams that will form. As observed in Fig. 8b, there is only one landslide (S.N.5) that too in the uncertain domain (between stability and instability). It implies that if it (S.N.5) forms a landslide dam, the dam might be stable or unstable depending upon the current river discharge, slope gradient, and landslide volume. Remaining landslides that are predicted to form the dams (SN. 7,14,15,19) belong to instability that means the resultant dams would not be durable. Similar explanations are mentioned in detail in Sec. 4.2 and 5.0.